# Estimation of the non-exceedance probability of extreme storm surges in South Korea using tidal-gauge data

Sang-Guk Yum,[1] Hsi-Hsien Wei,[2] Sung-Hwan Jang[3,4,*]

[1]      Department of Civil Engineering, Gangneung-Wonju National University, Gangneung, Gangwon-do 25457, South Korea; skyeom0401@gwnu.ac.kr

[2]      Department of Building and Real Estate, The Hong Kong Polytechnic University, Kowloon, Hong Kong, PR China; hhwei@polyu.edu.hk, hhweing@gmail.com

[3]      Department of Civil and Environmental Engineering, Hanyang University ERICA, Ansan, Gyeonggi-do 15588, South Korea; sj2527@hanyang.ac.kr

[4]      Department of Smart City Engineering, Hanyang University ERICA, Ansan, Gyeonggi-do 15588, South Korea; sj2527@hanyang.ac.kr

*_Correspondence to_: Sung-Hwan Jang (sj2527@hanyang.ac.kr)

**Abstract.** Global warming, one of the most serious aspects of climate change, can be expected to cause rising sea levels. These, in turn, have been linked to unprecedentedly large typhoons that can cause flooding of low-lying land, coastal invasion, seawater flows into rivers and groundwater, rising river levels, and aberrant tides. To prevent typhoon-related loss of life and property damage, it is crucial to accurately estimate storm-surge risk. This study therefore develops a statistical model for

estimating such surges' probability, based on surge data pertaining to Typhoon Maemi, which struck South Korea in 2003. Specifically, estimation of non-exceedance probability models of the typhoon-related storm surge was achieved via clustered separated peaks-over-threshold simulation, while various distribution models were fitted to the empirical data for investigating the risk of storm surges reaching particular heights. To explore the non-exceedance probability of extreme storm surges caused by typhoons, a threshold algorithm with clustering methodology was applied. To enhance the accuracy of such non-exceedance

probability, the surge data was separated into three different components: predicted water level, observed water level, and surge. Sea-level data from when Typhoon Maemi struck was collected from a tidal gauge station in the City of Busan, which is vulnerable to typhoon-related disasters due to its geographical characteristics. Fréchet, Gamma, log-normal, Generalised Pareto, and Weibull distributions were fitted to the empirical surge data, and the researchers compared each one's performance at explaining the non-exceedance probability. This established that Weibull distribution was better than any of the other

distributions for modeling Typhoon Maemi's peak total water level. Although this research was limited to one city in the Korean Peninsula and one extreme weather event, its approach could be used to reliably estimate non-exceedance probabilities in other regions where tidal gauge data are available. In practical terms, the findings of this study, and future ones adopting its methodology, will provide a useful reference for designers of coastal infrastructure.

# 1 Introduction

## 1.1 Climate change and global warming

Climate change, which can directly affect the atmosphere, oceans, and other planetary features via a variety of pathways and mechanisms, notably including global warming, also has secondary consequences for nature and for human society. In the specific case of global warming, one of the most profoundly negative of these secondary effects consists of sea-level rises, which can cause flooding of low-lying land, coastal invasion, seawater flows into rivers and groundwater, river-level rises, and tidal aberrations.

Recent research has also reported that, under the influence of global warming, the intensities and frequencies of typhoons and hurricanes are continuously changing, increasing these hazards' potential to negatively affect water resources, transport facilities, and other infrastructure, as well as natural systems (Noshadravan et al., 2017). Ke et al. (2018) studied these new frequencies of storm-induced flooding, with the aim of formulating new safety guidelines for flood-defence systems in Shanghai, China. They proposed a methodology for estimating new flooding frequencies, which involved analysing annual water-level data obtained from water-gauge stations along a river near Shanghai. The authors reported that a generalised extreme value (GEV) probability-distribution model was the best fit to the empirical data, and this led them to advocate changes in the recommended height of the city's flood wall. However, Ke at al. only considered annual maximum water levels when analysing flooding frequencies, which could have led to inaccurate estimation of the exceedance probability of extreme natural hazards such as mega-typhoons, which may bring unexpectedly or even unprecedentedly high water levels. In such circumstances, the protection of human society calls for highly accurate forecasting systems, especially as inaccurate estimation of the risk probability of these hazards can lead to the construction of facilities in inappropriate locations, thus wasting time and money as well as endangering life. Moreover, the combined effect of sea-level rises and tropical storms is potentially even more catastrophic than either of these hazards by itself.

## 1.1.1 Sea-level rises

According to the Intergovernmental Panel on Climate Change (IPCC, 2007), average global temperature increased by approximately 0.74℃ (i.e., at least 0.56℃ and up to 0.92℃) between 1906 and 2005 (Hwang, 2013). The IPCC's (2007) Fourth Assessment Report (AR4) noted that since 1961, world mean sea level (MSL) has increased by around 1.8mm (i.e., 1.3-2.3mm) per year; and when melting polar ice is taken into account, this figure increases to 3.1mm (2.4-3.8mm). Moreover, the overall area of Arctic ice has decreased by an average of 2.7% annually since 1978, and the amount of snow on mountains has also declined (Kim and Cho, 2013). These observations have sparked growing interest in how much sea levels will increase, including research on how changes in the climate can best be coped with (Radic and Hock, 2011; Schaeffer et al., 2012). Most industrial facilities on the Korean Peninsula, including plants, ports, roads, and shipyards, are located near the shore – as, indeed, are most residential buildings. These topographical characteristics make the cities of South Korea especially vulnerable to damage caused by sea-level rises and the associated large socioeconomic losses.

### 1.1.2 Sea-level rises potentially affecting the city of Busan, South Korea

Yoon and Kim (2012) investigated 51 years' worth of sea-level changes using data from tidal gauges at 17 stations located around the Korean Peninsula. They utilised regression analysis to calculate the general trend in MSL for 1960-2010 at each station, and found that around Korea, MSL rose more quickly than it did globally. The linear rising trend of MSL was relatively small along Korea's western coast (averaging 1.3mm/year), but large on the southern and eastern coasts (3.2mm/year and 2.0mm/year, respectively), and very large around Jeju Island (5.6mm/year, i.e., more than three times the global average). According to AR4, the rate of sea-level rises may accelerate after the 21st century, and this should be taken into consideration when designing coastal structures if disasters are to be avoided. Therefore, places most likely to be affected by current and future climate change need more accurate predictions of sea-level variation and surge heights, with a *surge* being defined as the difference between observed and predicted sea level. In the present work, Busan, a major metropolitan area on the south-eastern coast of South Korea, has been used as a case study. According to Yoon and Kim's (2012) calculations, the sea level around Busan rose by an average 1.8mm/year from 1960 to 2010, i.e., roughly the same as the global trend over the same period.

### 1.2 Problem statement

### 1.2.1 Typhoon trends in South Korea

The Korean Peninsula is bounded by three distinct sea-systems, generally known in English as the Yellow Sea, the Korea Strait, and the East Sea / Sea of Japan. This characteristic has often led to severe damage to its coastal regions. According to the Korea Ocean Observing and Forecasting System (KOOFS), Typhoon Maemi in September 2003 had a maximum wind speed of 54 metres per second (m/s), and these strong gusts caused an unexpected storm surge. This event caused US$3.5 billion in property damage, as shown in Table 1. All three of the highest peaks ever recorded by South Korea's tidal-gauge stations also occurred in that month.

Table 1. Largest typhoons to have struck the Korean Peninsula

| Name | Date | Amount of Damage (US$) | Max. Wind Speed (10 min. avg., m/s) |
|---|---|---|---|
| Rusa | 30 Aug.-1 Sep. 2002 | 4.3 billion (1st) | 41 |
| Maemi | 12-13 Sep. 2003 | 3.5 billion (2nd) | 54 |
| Bolaven | 25-30 Aug. 2012 | 0.9 billion (3rd) | 53 |

The most typhoon-heavy month in South Korea is August, followed by July and September, with two-thirds of all typhoons occurring in July and August. Tables 2 and 3, below, present statistics about typhoons in South Korea over periods of 68 years and 10 years ending in 2019, respectively; and Figure 1 shows the track of Typhoon Maemi from 4-16 September 2003. As can be seen from Figure 1, Typhoon Maemi passed into Busan from the southeast, causing direct damage upon landfall, after

which its maximum 10-minute sustained wind speed was 54 m/s. Typhoon Maemi prompted the insurance industry, the South Korean government, and many academic researchers to recognise the importance of advance planning and preparations for such storms, as well as for other types of natural disasters.

Table 2. Incidence of typhoons and typhoon landfall in South Korea, 1952-2019, by month

|  | Jan. | Feb. | Mar. | Apr. | May | Jun. | Jul. | Aug. | Sep. | Oct. | Nov. | Dec. | Total |
|---|---|---|---|---|---|---|---|---|---|---|---|---|---|
| Typhoons, $n$ | 29 | 15 | 15 | 45 | 67 | 115 | 245 | 351 | 322 | 238 | 152 | 73 | 1,678 |
| Typhoons, avg. | 0.54 | 0.28 | 0.46 | 0.83 | 1.24 | 2.13 | 4.54 | 6.52 | 5.96 | 4.41 | 2.81 | 1.35 | 31.07 |
| Landfalls, $n$ | 0 | 0 | 0 | 0 | 1 | 18 | 65 | 70 | 45 | 5 | 0 | 0 | 206 |
| Landfalls, avg. | 0.0 | 0.0 | 0.0 | 0.0 | 0.02 | 0.33 | 1.2 | 1.3 | 0.87 | 0.09 | 0.0 | 0.0 | 3.81 |

Table 3. Incidence of typhoons and typhoon landfall in South Korea, 2010-19, by month

|  | Jan. | Feb. | Mar. | Apr. | May | Jun. | Jul. | Aug. | Sep. | Oct. | Nov. | Dec. | Total |
|---|---|---|---|---|---|---|---|---|---|---|---|---|---|
| Typhoons, $n$ | 4 | 3 | 4 | 5 | 12 | 18 | 33 | 43 | 56 | 34 | 16 | 7 | 235 |
| Typhoons, avg. | 0.4 | 0.3 | 0.4 | 0.5 | 1.2 | 1.8 | 3.3 | 4.3 | 5.6 | 3.4 | 1.6 | 0.7 | 23.5 |
| Landfalls, $n$ | 0 | 0 | 0 | 0 | 0 | 0 | 3 | 11 | 7 | 5 | 2 | 0 | 28 |
| Landfalls, avg. | 0 | 0 | 0 | 0 | 0 | 0 | 0.3 | 1.1 | 0.7 | 0.5 | 0.2 | 0 | 2.8 |

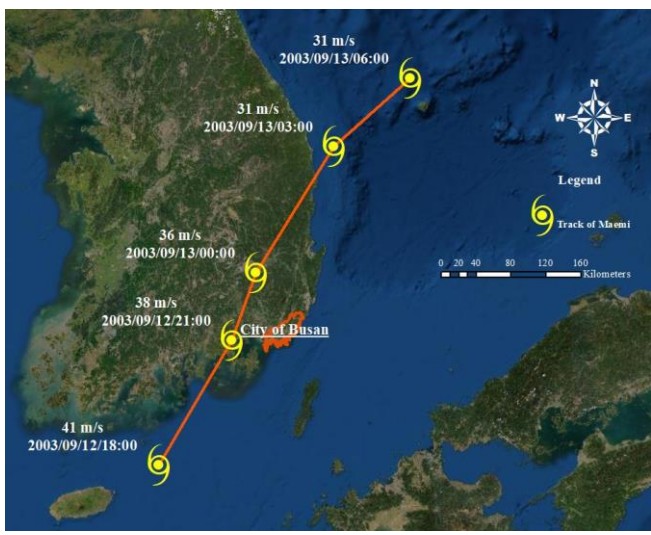

Figure 1. Track and wind speed of Maemi, 2003

**1.2.2 Tidal gauge stations in South Korea**

Effective measures for reducing the damage caused by future typhoons, including especially the design and re-design of waterfront infrastructure, will require accurate prediction of storm-surge height. When Typhoon Maemi struck the Korean Peninsula in 2003, South Korea was operating 17 tidal-gauge stations, of which eight had been collecting data for 30 years or more. They were located on the western (n=5), southern (n=10), and eastern coasts (n=2).

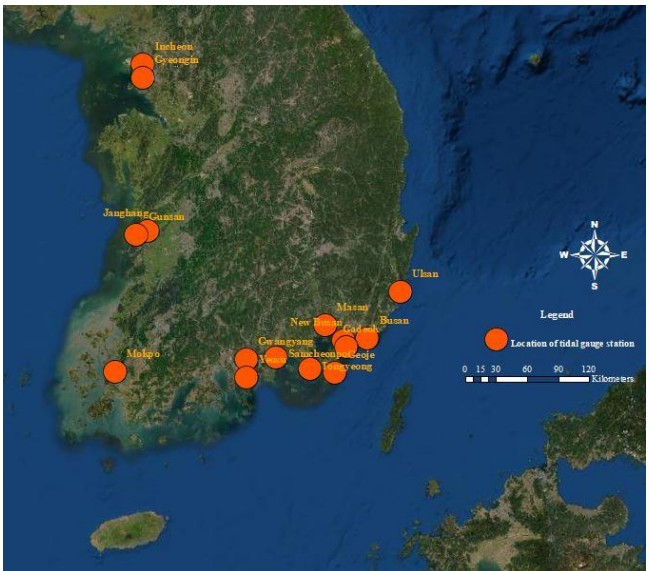

Figure 2. Locations of the 15 tidal-gauge stations on the western and southern coasts of South Korea as of 2003

This study focuses on the 15 tidal-gauge stations located on the southern and western coasts (Fig. 2). The reason for excluding the remaining two stations is that the majority of typhoons do not arrive from the east or make landfall on that coast. The hourly tidal data for this study has been provided by the Korea Hydrographic and Oceanographic Agency (KHOA) and is used with that agency's permission.

**1.2.3 Highest recorded water levels**

The western tidal-gauge stations are located at Incheon, Gyeongin, Changwon, Gunsan, and Mokpo. These five stations have operated for different lengths of time, ranging from 2 to 61 years. Collection of the sea levels observed hourly by each station throughout their respective periods of operation revealed the top three sea-level heights at each. These heights, which are shown in Table 4, are clearly correlated with the dates of arrival of typhoons.

Table 4. Three highest water levels recorded at each tidal-gauge station on Korea's west coast

| Location | Years of data | Top three peaks (cm) | Dates/times of peaks |
|----------|---------------|----------------------|----------------------|
| Incheon | 18 | 987 | 24 Jul. 2013, 10:00 |

| Location | Years of data | Top three peaks (cm) | Dates/times of peaks |
|----------|---------------|----------------------|----------------------|
| | | 981 | 8 Sep. 2002, 06:00 |
| | | 980 | 27 Oct. 2003, 18:00 |
| Gyeonin | 2 | 993 | 30 Sep. 2015, 19:00 |
| | | 987 | 29 Sep. 2015, 18:00 |
| | | 986 | 29 Oct. 2015, 18:00 |
| Janghang | 14 | 798 | 30 Sep. 2015, 17:00 |
| | | 796 | 11 Oct. 2014, 17:00 |
| | | 794 | 29 Sep. 2015, 16:00 |
| Gunsan | 37 | 805 | 19 Aug. 1997, 04:00 |
| | | 799 | 21 Aug. 1997, 05:00 |
| | | 797 | 31 Aug. 2000, 05:00 |
| Mokpo | 61 | 544 | 4 Jul. 2004, 04:00 |
| | | 544 | 6 Jul. 2004, 05:00 |
| | | 538 | 16 Nov. 2012, 16:00 |

The same approach was applied to the data from the 10 tidal-gauge stations on the south coast, as shown in Table 5 and Table 6, below.

Table 5. Three highest water levels recorded at nine of the 10 tidal-gauge stations on Korea's south coast

| Location | Years of data | Top three peaks (cm) | Dates/times of peaks |
|----------|---------------|----------------------|----------------------|
| Port of New Busan | 5 | 221 | 18 Sep. 2012, 10:00 |
| | | 219 | 17 Sep. 2012, 09:00 |
| | | 219 | 11 Aug. 2014, 21:00 |
| Gadeok | 40 | 252 | 17 Sep. 2012, 10:00 |
| | | 246 | 17 Sep. 2012, 09:00 |
| | | 246 | 16 Jul. 1987, 00:00 |
| Masan | 37 | 265 | 17 Sep. 2012, 10:00 |
| | | 264 | 17 Sep. 2012, 11:00 |
| | | 244 | 29 Aug. 2004, 21:00 |
| Ulsan | 55 | 133 | 19 Aug. 2004, 08:00 |
| | | 120 | 12 Sep. 2003, 21:00 |
| | | 129 | 17 Sep. 2012, 20:00 |
| Tongyeong | 41 | 426 | 12 Sep. 2003, 21:00 |
| | | 357 | 12 Sep. 2012, 10:00 |
| | | 356 | 12 Sep. 2003, 20:00 |

| | | | |
|---|---|---|---|
| Samcheonpo | 2 | 352 | 30 Aug. 2015, 22:00 |
| | | 350 | 28 Oct. 2015, 09:00 |
| | | 350 | 27 Nov. 2015, 10:00 |
| Geoje | 11 | 270 | 17 Sep. 2012, 09:00 |
| | | 259 | 17 Sep. 2012, 10:00 |
| | | 255 | 4 Jan. 2006, 09:00 |
| Gwangyang | 6 | 479 | 17 Sep. 2012, 10:00 |
| | | 443 | 17 Sep. 2012, 11:00 |
| | | 441 | 1 Aug. 2014, 22:00 |
| Yeosu | 52 | 440 | 18 Aug. 1966, 23:00 |
| | | 430 | 14 Sep. 1966, 21:00 |
| | | 129 | 17 Aug. 1966, 22:00 |

### 1.2.4 Tidal-gauge station at the City of Busan in South Korea

One of the focal tidal-gauge stations has observation records covering more than half a century. It is located on the south coast in Busan, South Korea's second-largest city. Thanks to its location near sea, Busan's international trade has boomed, and as a consequence it now boasts the largest port in South Korea. The Nakdong, longest and widest river in South Korea, also passes through it. Due to these geographical characteristics, Busan has been very vulnerable to natural disasters, and the importance of accurately predicting the characteristics of future storms is increasingly recognised by its government and other stakeholders.

The top three sea-level heights at the tidal-gauge station there are shown in Table 6, below. As this table indicates, all of the top-three water heights recorded in the long history of this station occurred during Typhoon Maemi's passage through the area.

Table 6. Three highest water levels recorded at the tidal-gauge station in Busan, South Korea

| | Years of data | Top three peaks (cm) | Dates/times of peaks |
|---|---|---|---|
| Busan | 54 | 211 | 12 Sep. 2003, 21:00 |
| | | 190 | 12 Sep. 2003, 20:00 |
| | | 188 | 12 Sep. 2003, 12:00 |

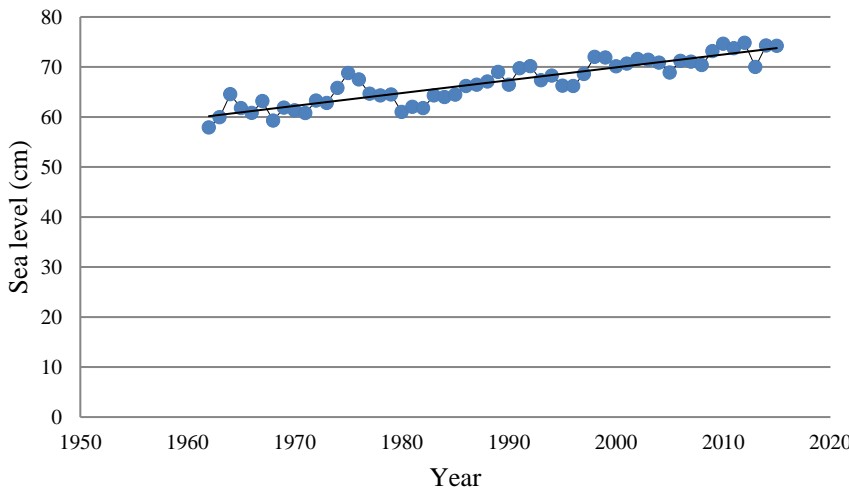

Figure 3. Mean Sea-level fluctuations in Busan, South Korea, 1962-2019 (Korea Hydrographic and Oceanographic Agency)


KHOA makes hourly observations of water height at the Busan tidal-gauge station, and the annual means presented in this paper have been calculated from that hourly data. As can be seen in Figure 3, plotting MSL for each year confirms that short-term water-level variation merely masks the long-term trend of sea-level increase. Therefore, on the assumption that MSL variation was a function of time, a linear regression was performed, with the resulting coefficient of slope indicating the rate

of increase (Yoon and Kim, 2012). The data utilised to estimate MSL for the tidal gauge station in Busan was provided by KHOA, which performed quality control on it before releasing it to us. Additionally, however, a normality test was performed, and the results (as shown in Table 7) indicated that the hourly sea-level data followed a normal distribution, at a significance >0.05. The Kolmogorov-Smirnov normality test was adopted as being well-suited to datasets containing more than 30 items.

Table 7. Kolmogorov-Smirnov normality test of sea-level fluctuation data from Busan tidal-gauge station

| | Statistic | df | Significance | Pearson correlation |
|---|---|---|---|---|
| Sea level fluctuation | 0.084 | 473352 | 0.200 | 0.96 |

As can be seen in Figure 3, the average rate of increase in MSL at Busan's tidal-gauge station from 1962 to 2019 was 2.4mm per year, yielding a difference of 16.31cm between the end of that period and the beginning. This finding is broadly in line with Yoon and Kim's (2012), that the rate of MSL increase around the Korean Peninsula as a whole between 1960 and 2010 was about 2.9mm/year. Also, linear regression analysis of the sea-level fluctuation data for 1965-2019 was utilised to discern the MSL trend. The significance level of 0.000 (<0.05) obtained via analysis of variance (ANOVA; Table 8) indicates that the

regression model of sea-level fluctuations was significant. Also, its correlation coefficient (0.96) indicated a strong positive relationship between sea-level rise and recentness. The coefficient of determination ($R^2$) was utilised to describe how well the

model explained the collected data. The closer $R^2$ is to 1, the better the model can predict the linear trend; and here, it was 0.74, as shown in Table 9. This means that the linear-regression model explained 74% of the sea-level variation. While this result suggests that the linear-regression analysis for sea-level fluctuation at the tidal gauge station in Busan is reliable, however,

such results may not be generalisable because variation in the data could have been due to several factors, including geological variation and modification of gauge points.

Table 8. Linear-regression coefficients, sea-level fluctuations at Busan tidal-gauge station

| | Non-standardised Coefficients | | Standardised Coefficients | t | Significance Probability (*P*-value) |
|---|---|---|---|---|---|
| | B | Standard Error | Beta | | |
| (Constant) | -422.23 | 35.022 | 0.887 | -12.06 | 0.00 |
| Sea level fluctuations | 0.246 | 0.018 | | 13.97 | 0.00 |

Table 9. Summary of analysis of variance results, sea-level fluctuations at Busan tidal-gauge station

| Model | Sum of Squares | df | Mean squares | F | Sig. | Adjusted $R^2$ |
|---|---|---|---|---|---|---|
| Regression | 830446354.04 | 41 | 20254789.12 | 32109.38 | .000b | 0.74 |
| Residual | 298566787.86 | 473310 | 630.81 | | | |
| Total | 1129013141.90 | 473351 | | | | |


### 1.2.5 Relationship between sea level and typhoons

When a storm occurs, surge height tends to increase, and these larger surges can cause natural disasters such as floods. In this study, before calculating the height of a surge, we took account of the dates and times when the three greatest sea-level heights were observed, as well the dates and times when typhoons occurred. These data are presented side by side in Table 10.

Table 10. Relation between sea levels and typhoons on the south coast of Korea

| Location | Years | Peak (cm) | Date | Typhoon |
|---|---|---|---|---|
| Busan | 54 | 211 | 12 Sep. 2003, 21:00 | Maemi |
| | | 190 | 12 Sep. 2003, 20:00 | Maemi |
| | | 188 | 12 Sep. 2003, 12:00 | Maemi |
| Port of New Busan | 5 | 221 | 18 Sep. 2012, 10:00 | Sanba |
| | | 219 | 17 Sep. 2012, 09:00 | Sanba |
| | | 215 | 18 Sep. 2012, 22:00 | Sanba |

| | | 252 | 17 Sep. 2012, 10:00 | Sanba |
|---|---|---|---|---|
| Gadeok | 40 | 246 | 17 Sep. 2012, 09:00 | Sanba |
| | | 246 | 16 Jul. 1987, 00:00 | Thelma |
| | | 265 | 17 Sep. 2012, 10:00 | Sanba |
| Masan | 37 | 264 | 17 Sep. 2012, 11:00 | n/a |
| | | 244 | 29 Aug. 2004, 21:00 | n/a |
| | | 133 | 19 Aug. 2004, 08:00 | Megi |
| Ulsan | 55 | 120 | 12 Sep. 2003, 21:00 | Maemi |
| | | 129 | 17 Sep. 2012, 20:00 | Sanba |
| | | 426 | 12 Sep. 2003, 21:00 | Maemi |
| Tongyeong | 41 | 357 | 12 Sep. 2012, 10:00 | Sanba |
| | | 356 | 12 Sep. 2003, 20:00 | Maemi |
| | | 352 | 30 Aug. 2015, 22:00 | n/a |
| Samcheonpo | 2 | 350 | 28 Oct. 2015, 09:00 | n/a |
| | | 350 | 27 Nov. 2015, 10:00 | n/a |
| | | 270 | 17 Sep. 2012, 09:00 | Sanba |
| Geoje | 11 | 259 | 17 Sep. 2012, 10:00 | Sanba |
| | | 255 | 4 Jan. 2006, 09:00 | n/a |
| | | 479 | 17 Sep. 2012, 10:00 | Sanba |
| Gwangyang | 6 | 443 | 17 Sep. 2012, 11:00 | Sanba |
| | | 441 | 1 Aug. 2014, 22:00 | n/a |
| | | 440 | 18 Aug. 1966, 23:00 | n/a |
| Yeosu | 52 | 430 | 14 Sep. 1966, 21:00 | n/a |
| | | 129 | 17 Aug. 1966, 22:00 | n/a |

As Table 10 indicates, the top three recorded sea levels at each south coast tidal-gauge station corresponded with the occurrence of typhoons in 20 out of 30 cases. Moreover, the dates and times of the three highest sea levels observed during all 57 years' worth of data from Busan all coincided with Typhoon Maemi passing out of the area.

As well as US$3.5 billion in property damage, Typhoon Maemi caused 135 casualties in Busan and nearby cities (National Typhoon Center, 2011). However, other typhoons – notably including Thelma, Samba, and Megi – also caused very considerable damage, as shown in Table 1.

## 2 Literature review

### 2.1 Prior studies of Typhoon Maemi

Most previous studies devoted to avoiding or reducing natural-disaster damage in South Korea have focused on storm characteristics, such as storm track, rainfall, radius, and wind-field data. Their typical approach has been to create synthetic storms that can be utilised to predict real storms' paths and estimate the extent of the damage they would cause.

Kang (2005) investigated the inundation and overflow caused by Typhoon Maemi at one location near the coast, using a site survey and interviews with residents, and found that the storm surge increased water levels by 80%. Using a numerical model, Hur et al. (2006) estimated storm surges at several points in the Busan area caused by the most serious typhoons, including Sarah, Thelma, and Maemi. Having established that Maemi was accompanied by the highest storm surge, they then simulated storm surges as a means of investigating the tidal characteristics of Busan's coast, and created virtual typhoons to compare against the actual tracks of Sarah, Thelma, and Maemi. When these virtual typhoons followed the track of Typhoon Maemi, their simulated storm surges were higher than the ones produced by those that followed the other two tracks.

Lee et al. (2008), using atmospheric-pressure and wind profiles of Typhoon Maemi, introduced a multi-nesting grid model to simulate storm surges. To check its performance, they used numerical methods for tidal calibration and to assess the influence of open-boundary conditions and typhoon paths. This yielded two key findings. First, the location of a typhoon's centre was the most critical factor when calculating storm surges; and second, the track of the typhoon was a secondary, but still important, factor in storm-surge prediction. However, Lee et al.'s research was limited by the fact that only recorded storm tracks were used, meaning that their simulations could not calculate storm surges from any other possible tracks. Similarly, Chun et al. (2008) simulated the storm surge of Typhoon Maemi used a numerical model, combined with moving boundary conditions to explain wave run-up, but using data from the coastal area of Masan: a city near Busan that was also damaged by the storm. The inundation area and depth predicted by Chun et al.'s model were reasonably well correlated with the actual area and depth arrived at via a site survey. Lastly, Kim and Suh (2018) created 25,000 random storms by modifying an automatic storm-generation tool, the Tropical Cyclone Risk Model, and then simulated surge elevations for each of them. The tracks of these simulated storms had similar patterns to those of actual typhoons in South Korea.

However, while past research on Typhoon Maemi has used such input data as tidal-gauge data, atmospheric pressure, wind fields, typhoon radius, storm speed, latitude, and longitude, tidal-gauge data has not been used for estimating the exceedance probabilities of storm surges. For instance, Kim and Suh (2018) did not perform surge modelling or frequency analysis in the time domain; and, although Chun et al.'s (2008) numerical models provided valuable predictions of inundation area and depth, they did not take account of tidal fluctuation which, if combined with increased water levels, would have yielded different results.

Using insurance data from when Typhoon Maemi made landfall on the Korean Peninsula, Yum et al. (2021) presented vulnerability functions linked to typhoon-induced high wind speeds. Specifically, the authors used insurance data to calculate separate damage ratios for residential, commercial, and industrial buildings, and four damage states adopted from an insurance

company and a government agency to construct vulnerability curves. Mean squared error and maximum likelihood estimation (MLE) were used to ascertain which curves most reliably explained the exceedance probability of the damage linked to particular wind speeds. Making novel use of a binomial method based on MLE, which is usually used to determine the extent of earthquake damage, the same study found that such an approach explained the extent of the damage caused by high winds in the Korean Peninsula more reliably than other existing methods such as theoretical probability method.

## 2.2 Return period estimates for Hurricane Sandy

While no prior research has estimated return periods for typhoons, some has done so for hurricanes. For example, Talke et al. (2014) used tidal-gauge data to study the storm-surge hazard in New York Harbor over a 37-year period, and found that its pattern underwent long-term changes due to sea-level rises caused in part by climate change. However, Talke et al. did not estimate a specific return period for Hurricane Sandy, which struck the United States in 2012.

Lin et al. (2010), on the other hand, did estimate the return periods of storm surges related to tropical cyclones in the New York City area, with that for Sandy in Lower Manhattan being 500 years within a 95% confidence interval (CI), i.e., approximately 400-700 years. Lin et al. (2012) later conducted a similar analysis using computational fluid dynamics Monte Carlo simulations that took account of the randomness of the tidal-phase angle. This approach yielded a return period of 1,000 years with a 90% CI (750-1,050 years). The former study can be considered the less accurate of the two, because it did not consider different surge-height possibilities at different time windows within the tidal cycle.

Hall and Sobel (2013) developed an alternative method to estimate Sandy return periods, based on the insight that this storm's track could have been the primary reason for the damage it caused in Lower Manhattan and other parts of the city. Specifically, they argued that Sandy's perpendicular impact angle with respect to the shore as it passed to the south of Manhattan's port was of critical importance, based on an analysis of the tracks of other hurricanes of similar intensity. They estimated the return period for Sandy's water level to be 714 years within a 95% CI (435-1,429 years).

Zervas (2013) estimated the return periods for extreme events using monthly mean water-level data from the U.S. National Oceanographic and Atmospheric Administration, recorded at the tidal-gauge station in Battery Park, New York. Using GEV distribution and MLE, Zervas calculated that the return period for Sandy's peak water level was 3,500 years; but sensitivity analysis suggested that the estimated results were probably inaccurate, given the GEV fit's sensitivity to the range of years used. Once Sandy was excluded, the return period was 60,000 years. This difference in results suggests that GEV distribution of the yearly maximum water level is not a realistic method for estimating extreme events in the New York Harbor area.

Building on her own past research, Lopeman (2015) – the first researcher to estimate Sandy's return period using tidal-gauge data – proposed that a clustered separated peaks-over-threshold (POT) method (CSPS) should be used, and that tide fluctuation, surge, and sea-level rise should all be dealt with separately, because out of these three phenomena, only surge is truly random. This approach led Lopeman to calculate the return period as 103 years with a 95% CI (38-452 years).

Zhu et al. (2017) explored recovery plans pertaining to two New York City disasters, Hurricanes Irene and Sandy, using data-driven city-wide spatial modelling. They used resilience quantification and logistic modelling to delineate neighbourhood

tabulation areas, which were smaller units than other researchers had previously used, and which thus enabled the collection of more highly detailed data. They also introduced the concept of *loss of resilience* to reveal patterns of recovery from these two hurricanes, again based on their smaller spatial units. Moran's *I* was utilised to confirm that loss of resilience was strongly correlated not only with spatial characteristics, but also with socioeconomic ones, and factors like the location of transport systems. However, given the particularity of such factors, Zhu et al.'s results might not be generalisable beyond New York

City; and they made no attempt to predict future extreme events' severity or frequency.

The sharp differences in the results of the past studies cited above are due to wide variations in both the data they used and their assumptions. The present study therefore applies all of the methods used in previous studies of Hurricane Sandy's return period to estimate that of Typhoon Maemi, and in the process, establishes a new model.

## 2.3 Extreme-value statistics

### 2.3.1 Prior studies of extreme natural hazards

Bermúdez et al. (2019) studied flood drivers in coastal and riverine areas as part of their approach to quantifying flood hazards, using 2D shallow-water models to compute the correlation between extreme events and flood drivers. They also adopted ordinary least-squares regression analysis to construct a 10,000-year time series, and computed water levels' exceedance probabilities for comparison. However, the possibility of river discharges, sea-wave trends, and tidal fluctuations were not

considered in their study.

The wrecking of windfarms by extreme windstorms is of considerable concern in the North Sea region, which is home to 38 such farms belonging to five different countries. According to Buchana and McSharry's (2019) Monte Carlo simulation-based risk-management study, the total asset value of these windfarms is €35 billion. It used a log-logistic damage function and Weibull probability distribution to assess the risks posed to windfarms in that region by extreme strong winds, and exceedance

probability to predict the extent of financial loss from such damage, in terms of solvency capital requirement (SCR). The same study also simulated the results of various climate-change scenarios, and the results confirmed that higher wind speed and higher storm frequency were correlated with rises in SCR: a finding that could be expected to help emergency planners, investors, and insurers reduce their asset losses.

According to Catalano et al.'s (2019) study of high-impact extratropical cyclones (ETCs) on the north-eastern coast of the

Unites States, limited data caused by these storms' rarity made it difficult to predict the damage they would cause, or analyse their frequency. To overcome this, they utilised 1,505 years' worth of simulations derived from a long-coupled model, GFDL FLOR, to estimate these extreme events' exceedance probabilities, and compared the results against those of short-term time-series estimation. This revealed not only that the former was more useful for statistical analysis of ETCs' key characteristics – which they defined as maximum wind speed, lowest pressure, and surge height – but also that the use of a short time-series

risked biasing estimates of ETCs' return levels upwards (i.e., underestimating their actual frequency). While these results

regarding return levels and time-series were valuable, however, Catalano et al. did not distinguish between the cold season and the warm season of each year, which could also have led to biased results.

A joint-probability methodology was used to analyse extreme water heights and surges on China's coast by Chen et al. (2019). They obtained the sea-level data from nine gauge stations, and utilised 35 years' worth of simulation data with Gumbell
distribution and Gumbell-Hougaard copula. The three major sampling methods proposed in the study were structural-response, wave-dominated, and surge-dominated sampling. The first was utilised to assess structures' performance in response to waves and surges. Joint-probability analysis revealed that such performance was correlated with extreme weather events in the target region, and that such correlation became closer when wave motion was stronger. Also, based on their finding that joint exceedance probability tended to overestimate return periods for certain water levels, Chen et al. recommended that offshore
defence-facility designers use joint-probability density to estimate return levels of extreme wave heights. Yet, while their study provided a useful methodology, particularly with regard to sampling methods and probability modelling of return periods and structural performance, they only looked at China's coast, and therefore their findings are unlikely to be generalizable to the Korean Peninsula.

Davies et al. (2017) proposed a framework for probability modelling of coastal storm surges, especially during non-stationary
extreme storms, and tested it using the El Niño-Southern Oscillation (ENSO) on the east coast of Australia. Importantly, they applied their framework to ENSO and seasonality separately. This is because, while ENSO affects storm-wave direction, mean sea level, and storm frequency, seasonality is mostly related to storm-surge height, storm-surge duration, and total water height. This separation has the advantage of allowing all storm variables of non-stationary events to be modelled, regardless of their marginal distribution. Specifically, Davies et al. applied non-parametric distribution to storm-wave direction and steepness,
and parametric distribution to duration and surge using mixture-generalised extreme value probability modelling, which they argued was more useful than standard ones such as Generalized Pareto Distribution (GPD). This, they said, was because the statistical threshold in an extreme mixture model can be integrated into the analysis, whereas a GPD model should be given an unbiased threshold: if it is low, too many normal data may be included. Accordingly, they utilised bootstrapping for the confidence interval to show the uncertainty of the non-stationary aspects of the extreme events. Also, they added a Bayesian
method to provide wider confidence intervals with less bias. Their findings are mainly beneficial to overcoming the challenges of GPD threshold selection; however, robust testing of their approach will require that it be applied to a wider range of abnormal climate phenomena.

Similar research was conducted by Fawcett and Walshaw (2016), who developed a methodology for estimating the return levels of extreme events such as sea surges and high winds of particular speeds, with the wider aim of informing practical
applications such as design codes for coastal structures. They reported that two of the most popular existing methods for doing so, block maxima (BM) and POT, both have shortcomings, and concluded that a Bayesian approach would be more accurate. Specifically, they argued that BM and POT methods tend to waste valuable data, and that considering all exceedance via accurate estimation of the extremal index (reflecting uncertainty's natural behaviour) could compensate for this disadvantage. They further proposed the seasonal variations should be taken into consideration with the all exceedance data, where possible.

In response to Japanese government interest in unexpected flooding caused by extreme storm surges during typhoons and other high-wind events, Hisamatsu et al. (2020) simulated typhoons as a means of predicting the cost of the damage they would cause in Tokyo Bay, which is very vulnerable to such events due to its geographic and socio-economic characteristics. Using stochastic approaches, they modelled future typhoons over a 10,000-year period, and calculated flooding using a numerical surge model based on the probability of historical typhoons. These flooding calculations, in turn, were utilised to

create a storm-surge inundation map, representing exceedance probabilities derived from stochastic hazard calculations pertaining to 1,000 typhoons. Next, the completed map was overlaid on government-provided values of Tokyo Bay's buildings and other infrastructural elements, to assess the spatial extent and distribution of the likely damage. The results showed that Chiba and Kanagawa would be the most damaged areas, and suffer financial losses of ¥158.4 billion and ¥91.5 billion, respectively, with an exceedance probability of 0.005 (as commonly used to estimate damage in the insurance

industry). However, the real-estate values they used were two decades out of date at the time their study was conducted, meaning that further validation of their approach will be needed.

Another effort to estimate return periods was made by McInnes et al. (2016), who created a stochastic dataset on all cyclones that occurred near Samoa from 1969 to 2009. That dataset was utilized to model storm tides using an analytic cyclone model and a hydrodynamic model, which also took account of prevailing climate phenomena such as La Niña and El Niño when

estimating return periods. The authors found that tropical cyclones' tracks could be affected by La Niña and El Niño, and more specifically, that the frequency of cyclones and storm tides during El Niño was consistent across all seasons, whereas La Niña conditions make their frequency considerably lower in La Niña season. Additionally, McInnes et al. proposed that sea-level rises had a more significant influence on storm tides than on future tropical cyclones did, based on their finding that future cyclones' frequency would be reduced as the intensity of future cyclones increased. Lastly, they found that the

likelihood of a storm tide exceeding a 1% annual exceedance probability (i.e., a one-in-a-hundred year tide) was 6% along the entire coastline of Samoa. However, other effects such as sea level fluctuations and meteorological factors were not included in their calculations.

Silva-González et al. (2017) studied threshold estimation for analysis of extreme wave heights in the Gulf of Mexico, and argued that appropriate thresholds for this purpose should consider exceedances. They applied the Hill estimator method, an

automated threshold-selection method, and the square-error method for threshold estimation in hydrological, coastal engineering, and financial scenarios with very limited data, and found that the square-error method had the most advantages, because it did not consider any prior parameters that could affect thresholds. The authors went on to propose improvements to that method, i.e., the addition of differences between quantiles of the observed samples and median quantiles from GPD-aided simulation. When GPD was utilised to estimate observed samples, it effectively prevented convergence problems with the

maximum-likelihood method when only small amounts of data were available. The key advantage of Silva-González et al.'s approach is that the choice of a threshold can be made without reliance on any subjective criteria. Additionally, no particular choice of marginal probability distribution is required to estimate a threshold. However, to be of practical value, their method will need to incorporate more meteorological factors.

Lastly, Wahl et al.'s (2015) study of the exceedance probabilities of a large number of synthetic and a small number of actual storm-surge scenarios utilized four steps: parameterising the observed data; fitting different distribution models to the time series; Monte Carlo simulation; and recreating synthetic storm-surge scenarios. Specifically, projected 40cm and 80cm sea-level rises were used as the basis for investigating the effects of climate change on flooding in northern Germany. Realistic joint-exceedance probabilities were used for all parameters with copula models; and the exceedance probabilities of storm surges were obtained from the bivariate exceedance probability method with two parameters, i.e., the highest total water level with the tidal fluctuations and intensity. Wahl et al.'s findings indicated that extremely high water levels would cause substantial damage over a short time period, whereas relatively small storm surges could inflict similar levels of damage but over a much longer period. However, like various other studies cited above, Wahl et al.'s did not take account of seasonal variation.

### 2.3.2 Generalized extreme value distribution

Extreme events are hard to predict because data points are so few, and predicting their probability is particularly difficult due to their asymptotic nature. Extreme-value probability theory deals with how to find outlier information, such as maximum or minimum data values, during extreme situations. Examining the tail events in a probability distribution is very challenging. However, it is considered very important by civil engineers and insurers, due to their need to cope with low-probability, high-consequence events. For example, the designs and insurance policies of bridges, breakwaters, dams, and industrial plants located near shorelines or other flood-prone areas should account for the probability, however low, of major flooding. Various probability models for the study of extreme events could potentially be used in the present research, given that its main topic is the extreme high water levels caused by typhoons. Extreme-value theories can be divided into two groups, according to how they are defined. In the first, the entire interval of interest is divided into a number of subintervals. The maximum value from each subinterval is identified as the extreme value for it, and then the entirety of these extreme values converges into a GEV distribution. In the second group, values that exceed a certain threshold are identified as extreme, and converge to a GPD. The following two subsections discuss the BM and POT methods, as illustrations of these two groups, respectively (Coles, 2001).

*Block-maxima method*

The BM approach relies on the distribution of the maximum extreme values in the following equation,

$$M_n = \max\{X_1, \dots, X_n\} \qquad (1)$$

where the $X_n$ series, comprising independent and identically random variables, occurs in order of maximum extreme values; n is the number of observations in a year; and $M_n$ is the annual maximum.

Data is divided into blocks of specific time periods, with the highest values within each block collectively serving as a sample of extreme values. One limitation of this method is the possibility of losing important extreme-value data because only the single largest value in each block is accounted for, and thus, the second-largest datum in one block could be larger than the highest datum in another.

*Peaks-over-threshold method*

The POT method can address the above-mentioned limitations of BM, insofar as it can gather all the data points that exceed a certain prescribed threshold, and use limited data more efficiently because it relies on relatively larger or higher values instead of the largest or highest ones. All values above the threshold – known as exceedances – can be explained by the differentiated tail-data distribution. The basic function of this threshold is to assort the larger or higher values from all data, and the set of exceedances constitutes the sample of extreme values. This means that, although POT can capture potentially important extreme values even when they occur close to each other, selecting a threshold that will yield the best description of the extreme data can be challenging (Bommier, 2014). That is, if it is set too high, key extreme values might be lost, but if it is set too low, values that are not really extreme may be included unnecessarily. Determining appropriate threshold values thus tends to require significant trial and error, and various studies have proposed methods for optimizing such values (Lopeman et al., 2015; Pickands, 1975; Scarrott and Macdonald, 2012). Pickands (1975), for instance, suggested that independent time-series that exceed high-enough thresholds would follow GPD asymptotically, thus avoiding the inherent drawbacks of BM. The distribution function F of exceedance can be computed as

$$F_\theta(x) = P\{X - \theta \leq x | X > \theta\}, \quad x \geq 0 \tag{2}$$

where θ is the threshold and X is a random variable.

$F_u$, meanwhile, can be defined by conditional probabilities:

$$F_\theta(x) = \begin{cases} \frac{F(\theta + x) - F(\theta)}{1 - F(\theta)} & \text{if } x \geq 0 \\ 0 & \text{else} \end{cases} \tag{3}$$

According to Bommier (2014), the distribution of exceedances $(Y_1, \ldots, Y_{n_\theta})$ can be generalised by GPD with the following assumption: When $Y = X - \theta$ for $X > \theta$, and $X_1, \ldots, X_n$, $Y_j = X_i - \theta$ can be described with i, which is the jth exceedance, $i = 1, \ldots, n_\theta$.

The GPD can be expressed as

$$G_x(x; \xi, \sigma, \theta) = \begin{cases} 1 - \left(1 + \xi \frac{(x-\theta)}{\sigma}\right)^{1/\xi} & \xi \neq 0 \\ 1 - \exp\left(-\frac{(x-\theta)}{\sigma}\right) & \xi = 0 \end{cases} \tag{4}$$

with x being independent and identically random variables; σ, the scale; ξ, the shape; and θ, the threshold. All values above θ are considered tail data (extreme values). The probability of exceedance over a threshold when calculating a return level that is exceeded once every N years (N-year return periods $x_N$), is calculated as:

$$P\{X > x | X > \theta\} = \left[1 + \xi\left(\frac{x-\theta}{\sigma}\right)\right]^{-1/\xi} \tag{5}$$

If the exceedances above the threshold are rare events λ (as measured by number of observations per year), we can expect $P(X > \theta)$ to follow Poisson distribution. The mean of exceedance per unit of time (ŷ) describes that distribution.

$$P(X > \theta) = \frac{\gamma}{\lambda} \tag{6}$$

In other words, γ can be estimated by dividing the number of exceedances by the number of years in the observation period.

Combining the POT and Poisson processes with GPD allows us to describe the conditional probability of the extreme values that exceed the designated threshold, as per Eq. (7) (Lopeman et al., 2015):

$$P(A|B) = \frac{P(A \cap B)}{P(B)} \tag{7}$$

And, when Bayes' theorem is applied to the role of GPD in conditional probability, we can rewrite Equation (7) as follows:

$$G_X(x) = \frac{P(\theta < X < x)}{P(X > \theta)} \tag{8}$$

## 3 Research methods

The objective of this study is to estimate the probability of the risk, in years, of typhoon-induced high water levels in Busan. To that end, it adapts Lopeman et al.'s (2015) CSPS, which provides statistical analysis of extreme values in long time-series of natural phenomena. As such, CSPS can provide useful guidance to those tasked with preparing for natural disasters on the

Korean Peninsula, and perhaps especially on its southern coast. The findings from this research are therefore expected provide a viable method of predicting economic losses associated with typhoons, and corresponding models for managing emergency situations arising from natural disasters, that can be used by South Korea's government agencies, insurance companies, and construction industry. And, although this study focuses on a specific city-region, its proposed probabilistic methodologies should be applicable to other coastal regions in South Korea and around the world.

To explore the non-exceedance probability of storm surges, this study utilised tidal-gauge data from the city of Busan, collected when Typhoon Maemi struck it in 2003. As shown in Figure 4, we proceeded according to several steps. First, the observed tidal-gauge data was utilized to calculate the predicted water level through harmonic analysis, and then, the storm surge height, which is difference between observed and predicted water height. Second, threshold and clustering techniques were applied to select data meaningful to the non-exceedance probabilities of extreme storm surges. Third, the extreme values were separated

into cold-season and warm-season categories, to boost the reliability of our probability-distribution model. Fourth, the maximum likelihood method was used to estimate non-exceedance probability. And fifth, various probability models were built, and the one that best fit the empirical data identified.

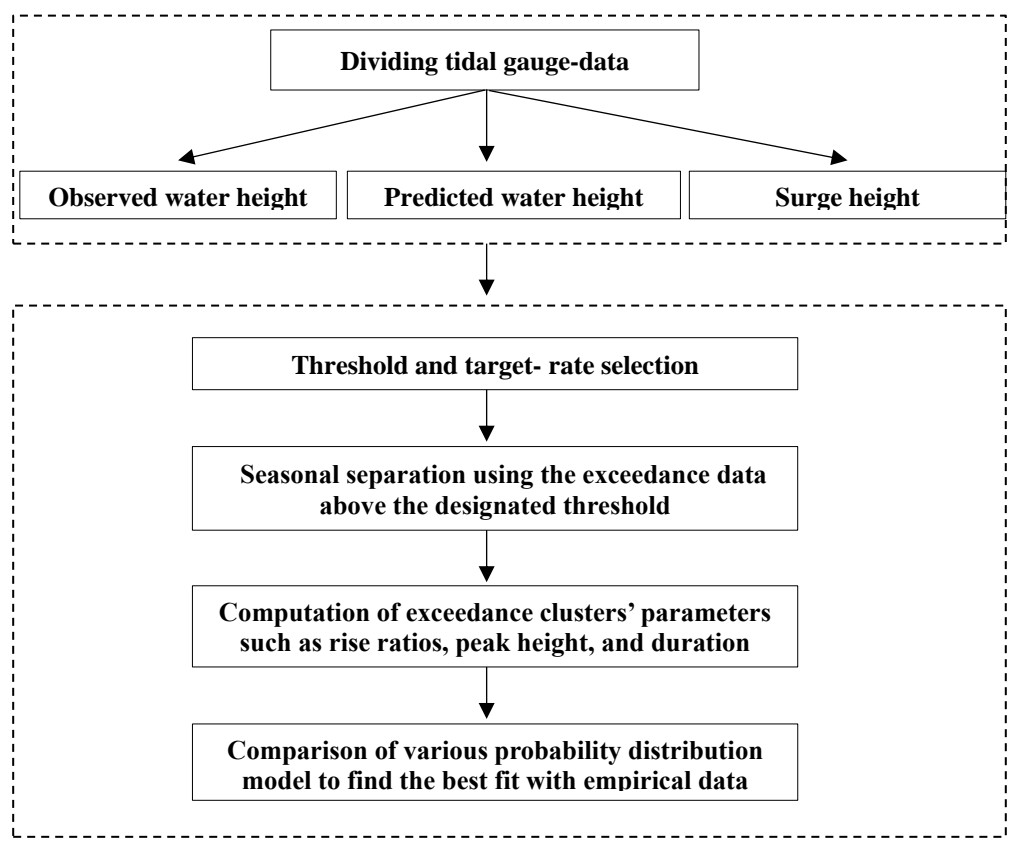

Figure 4. General approach and workflow

## 3.1 Data processing

### 3.1.1 Storm-surge data collection method

To determine the height of surges from publicly available KHOA data, it was first necessary to predict sea levels. Eq. (9) explains the interrelationship of observed water level, predicted water level, tidal-fluctuation height, and residual (surge) at time $t_i$,

$$Y_i = X_i + S_i \tag{9}$$

where $i = 1, 2, \ldots, n$; n is the time series of the input dataset; $X_i$, the predicted water height at $t_i$; $Y_i$, the observed water height at $t_i$; and $S_i$, the surge height.

### 3.1.2 Separation of tidal gauge data via harmonic analysis

A standard harmonic analysis was performed to calculate predicted sea-level height based on hourly tidal-gauge data. First, this technique was used to estimate the tidal components of all seawater-level data, allowing residuals to be isolated so that surge data could be calculated once sea-level rise had been estimated. Second, the estimated constituents were used to predict

tidal fluctuations in the years simulated via Monte Carlo. Then, the TideHarmonics package in R (Stephenson, 2017) was used to estimate tidal components, as detailed below.

Given a time-series Y(t) of total water levels, with t denoting time in hours, the tidal component with M harmonic constituents is computed as

$$\hat{Y}(t) = Z + \sum_{m=1}^{M} A_m cos\left(\frac{\pi}{180}(\omega_m t - \psi_m)\right) \tag{10}$$

where $\omega_m$ is the angular frequency of the m-th component in degrees per hour. The $2M + 1$ parameters to be estimated are the amplitudes $A_m$; the phase lags $\psi_m$ in degrees; and the MSL, Z.

To account for long astronomical cycles (LAC), nodal-correction functions for both the amplitude and phase are used. With these corrections, the tidal component takes the form

$$\hat{Y}_{LAC}(t) = Z + \sum_{m=1}^{M} H_m f_m(t) cos\left(\frac{\pi}{180}(\omega_m t - g_m + u_m(t) + V_m)\right) \tag{11}$$

where $f_m(t)$ and $u_m(t)$ respectively represent the nodal corrections for the amplitude and phase. In this new formulation, the amplitude and phase parameters to be estimated are denoted by $H_m$ and $g_m$ (in degrees). Finally, $V_m$ is the reference signal, by which the phase-lag $g_m$ is calculated and set to refer to the origin $t = 0$.

The summation term in $\hat{Y}_{LAC}(t)$ can alternatively be written as

$$\sum_{m=1}^{M} \beta_{m,1} f_m(t) \cdot cos\left(\frac{\pi}{180}(\omega_m t + u_m(t) + V_m)\right) + \sum_{m=1}^{M} \beta_{m,2} f_m(t) \cdot sin\left(\frac{\pi}{180}(\omega_m t + u_m(t) + V_m)\right) \tag{12}$$

where $\beta_{m,1} = H_m cos(g_m)$ and $\beta_{m,2} = H_m sin(g_m)$. What is gained from this new representation is a linear function with respect to the parameters $\beta_{m,1}$ and $\beta_{m,2}$ that need to be estimated; and hence, linear regression can be used. Given the large timespan covered by the data, $M = 60$ harmonic tidal constituents were estimated, and a constant mean sea level Z was assumed across all years of available data.

### 3.1.3 Observed, predicted and residual water levels

Because observed sea level usually differs from predicted sea level, Figure 5 depicts the former (as calculated through harmonic analysis) in blue. Predicted sea levels are shown in green, and surge height in red. As the figure indicates, the highest overall water level coincided with the highest surge during Typhoon Maemi, i.e., at 21:00 on 12 September 2003. Given a total water height of 211cm, the surge height was calculated as 73.35cm. The unexpectedly large height of the surge induced by Typhoon Maemi caused US$3.5 billion in property damage and many causalities in Busan, as mentioned in Table 1.

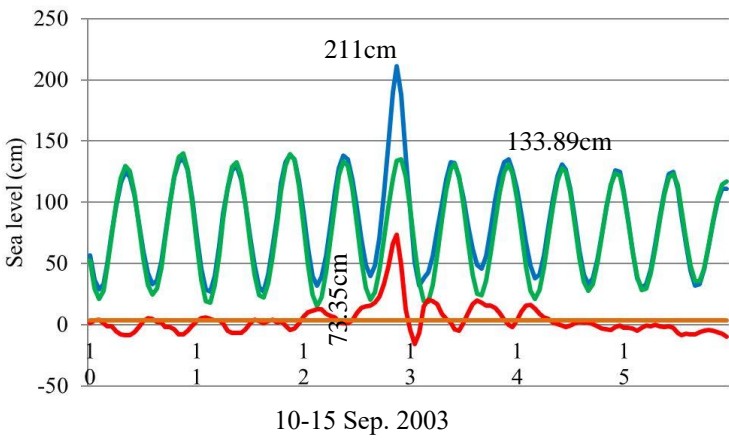

Figure 5. Observed (green), predicted (blue), and residual (red) water levels at Busan during Typhoon Maemi

**3.2 Data analysis**

**3.2.1 Threshold and target-rate selection**

At a given annual target rate – i.e., number of storms per year – the algorithm proposed by Lopeman et al. (2015) (Fig. 6) computes the threshold such that this rate approximates the resulting yearly number of *exceedance clusters*, i.e., consecutive surge observations that lie above the threshold. Hence, rather than choosing an "ideal" threshold according to some criterion or other, the algorithm simply finds the threshold that forces a chosen target rate to occur. Accordingly, a study of this kind could set its target rate as the average rate observed over a given period, or as a value that the researchers find reasonable in light of their knowledge of historical data for their focal area.

Next, the algorithm iteratively updates the threshold to allow a computationally intensive, but not exhaustive, exploration of possible threshold values between its minimum value (i.e., here, minimum observed surge height) and its maximum one (i.e., maximum observed surge height). Specifically, it first sets the threshold to 0cm, and then iteratively overwrites it according to the following steps.

(1) The exceedance clusters produced at a given iteration and given threshold are identified, and the resulting annual storm rate computed.

(2) If the annual storm rate arrived at in step (1) is equal to, or about equal to, the chosen target, then the threshold from the previous iteration is the result, and the algorithm is stopped.

(3) If the annual storm rate arrived at in step (1) is not close to the chosen target, but…

    (a) …is smaller than the target rate, then the threshold from the previous iteration is the result, and the algorithm is stopped; or

(b) …is larger than the target rate, then a vector collecting the maximum height of the clusters is built and sorted in descending order. The threshold is then updated by setting it as equal to the C-th element of this vector, where C is the integer closest to 54 (i.e., the number of years covered by the dataset) multiplied by the target rate. This updated threshold is used in the next iteration of the algorithm, and steps (1) through (3) are repeated.

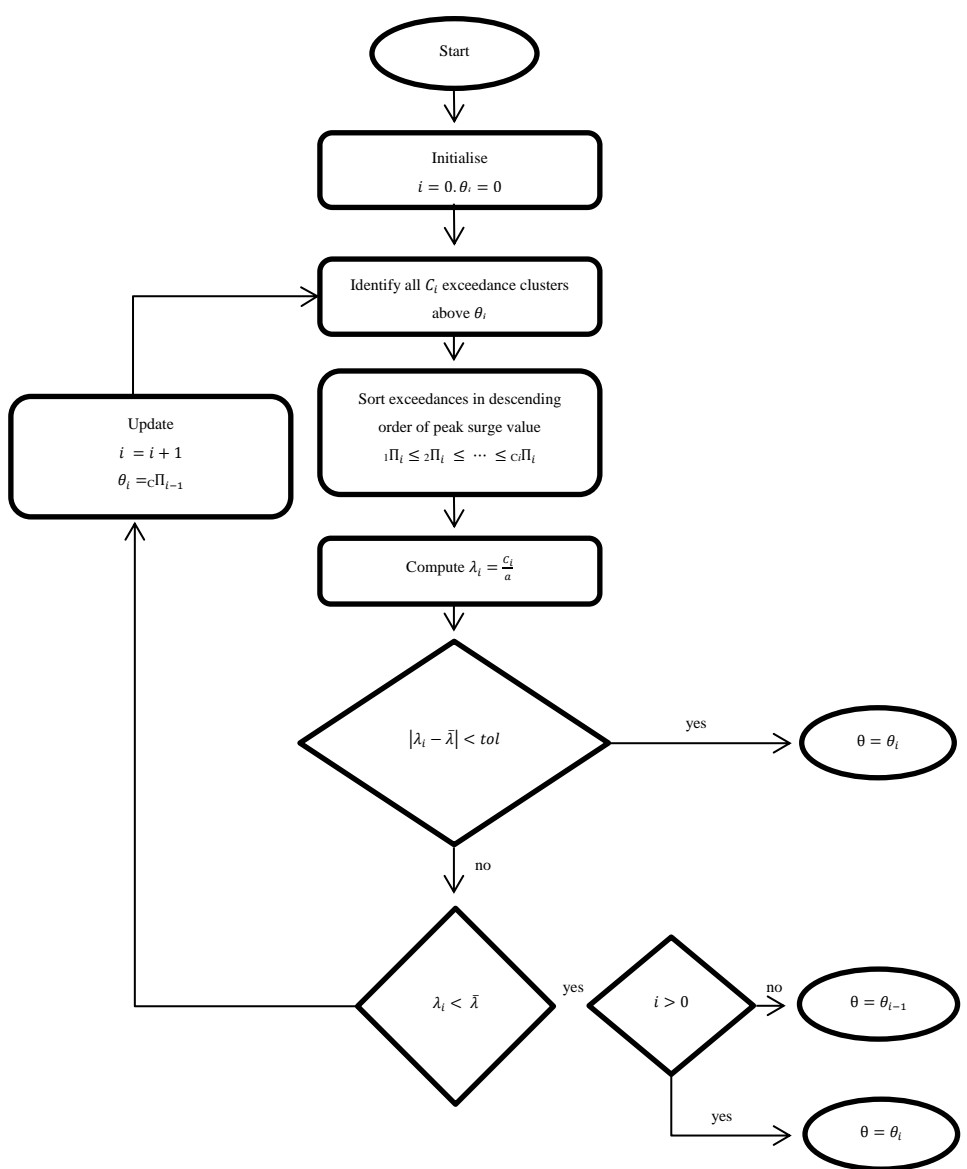

Figure 6. Threshold-selection flowchart

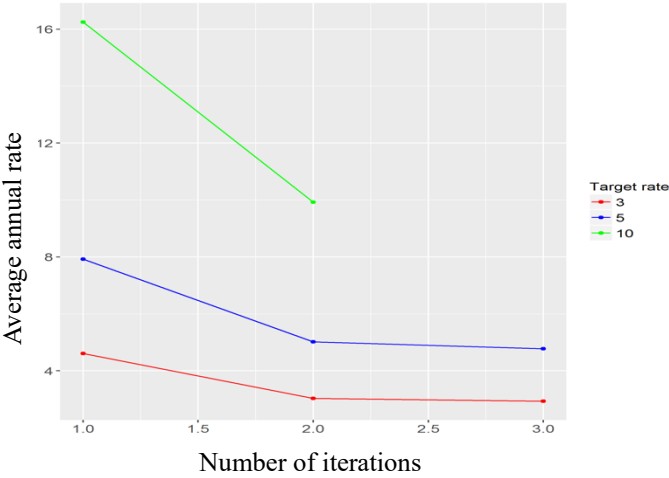

Figure 7. Iterative process of threshold selection (1 of 3)

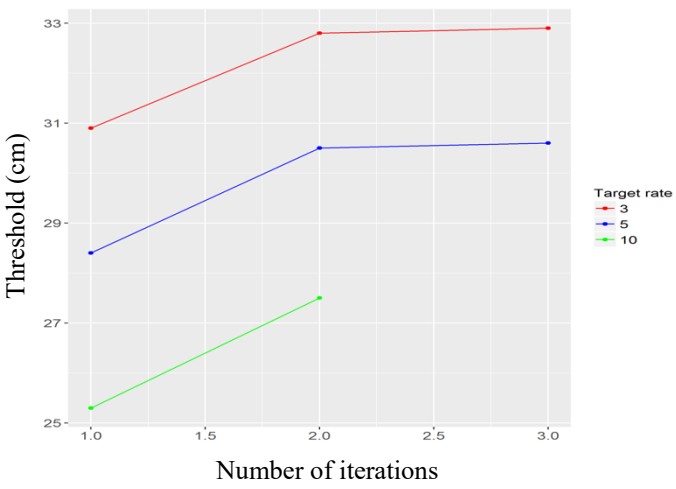


Figure 8. Iterative process of threshold selection (2 of 3)

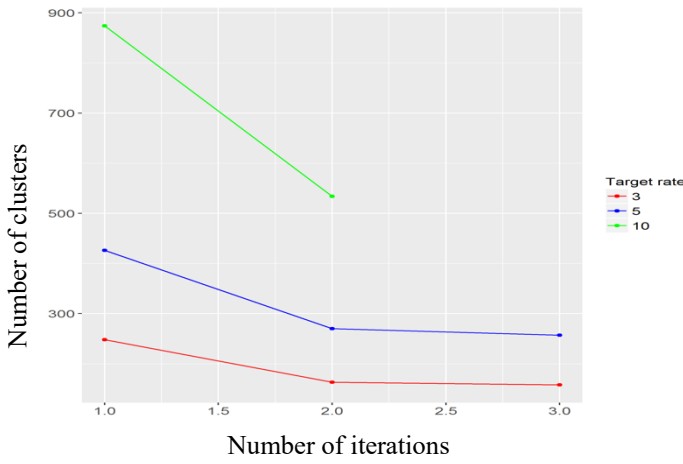

Figure 9. Iterative process of threshold selection (3 of 3)

As shown in Figures 7, 8, and 9, the threshold algorithm (Fig. 6) achieved convergence relatively quickly for all three target
rates selected, with the number of iterations required for convergence ranging from three (with a target rate of 3.0) to five (with
a target rate of 10).

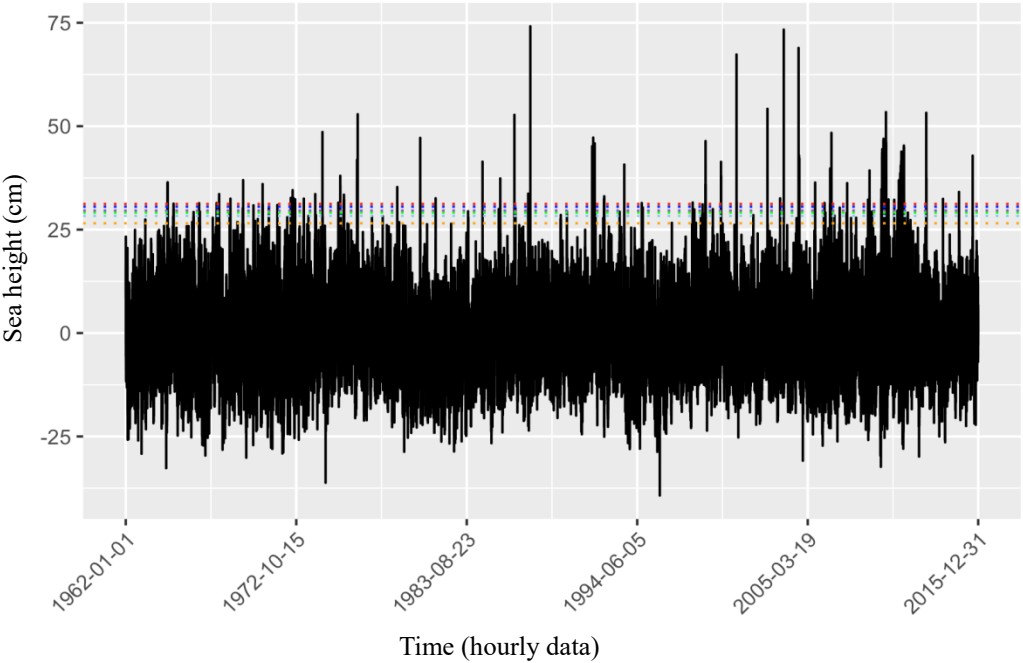

Figure 10. Various thresholds considered

Figure 10 displays six possible thresholds. The first, of 31.2cm, was based on a target rate of 3.5, and 189 clusters, and is shown in red. The dark-blue line represents the second threshold, of 30.54cm, (target rate=4.0, clusters=217); the purple, a threshold of 29.56cm (target rate=4.5, clusters=246); the green, a threshold of 29.15cm (target rate=5.0, clusters=274); the sky-blue, a threshold of 28.33cm (target rate=6.0, clusters=324); and the orange, a threshold of 26.53cm (target rate=8.0, clusters=431).

### 515 3.2.2 Clustering of the storm-surge data: Interrelationship of target rate, threshold and clusters

Figures 7, 8, and 9 show that, as expected, when the target rate increases, the threshold decreases, and as the threshold decreases, the number of clusters (i.e., storm events) increases. Conversely, the lower the target rate, the lower the number of clusters and the higher the threshold. Thus, if the desired number of storms is three per year, the algorithm will converge in three iterations and set the threshold level to 32.01cm; this results in a total of 164 storm events over the timespan covered by our data.
Conversely, if the desired target rate is 10 storms per year, the threshold is significantly lower (25.43cm), and the total number of storm events more than trebles, to 539 clusters (Figure 10). As can be seen in Figure 11, we chose only one maximum value to represent each cluster. Figures 12, 13, and 14 show the stages of the clustering of surges when the target rate is set to 5.0, the threshold is 29.15cm, and the number of clusters is 274 (though it should be noted that Figure 12 indicates only the number of surges, due to the difficulty of visually representing all surge dates and times from the period 1962-2019). The surge data
above the designated threshold, arrived at via the threshold-selection method above, are shown in Figure 12. Here, the data above the threshold are clustered based on their start and end times, and again, just one maximum value was chosen at each cluster. Figure 14 presents all maxima obtained from a cluster separately.

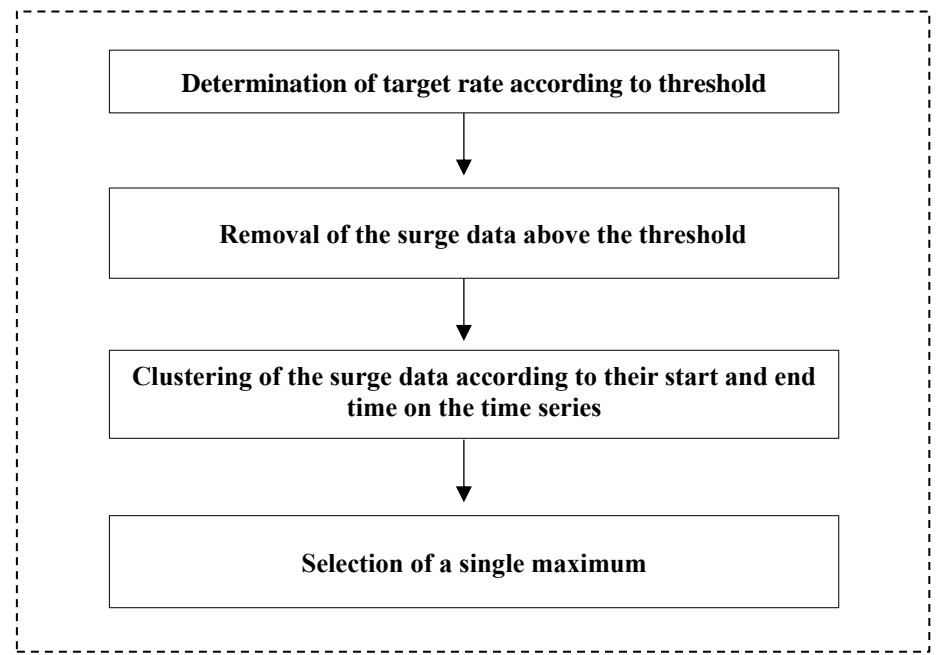

Figure 11. Clustering flowchart


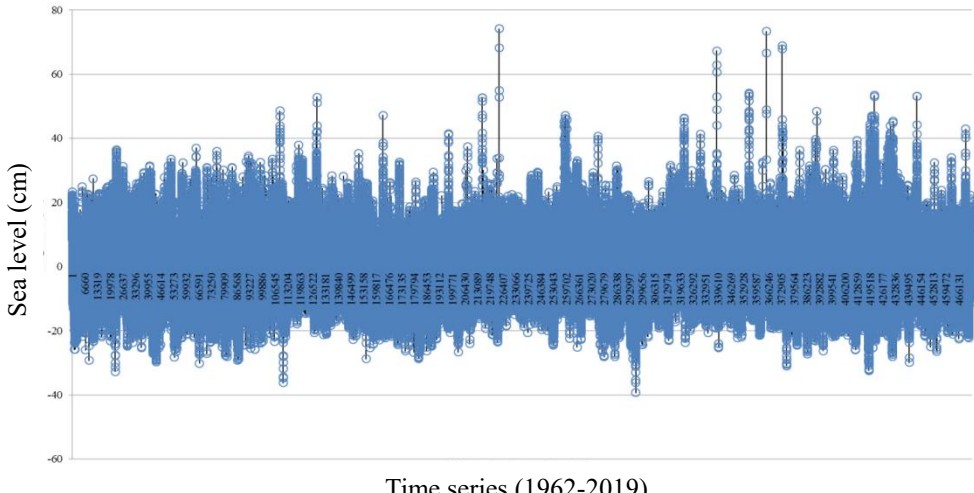

Figure 12. Data from Busan tidal-gauge station, before application of any thresholds

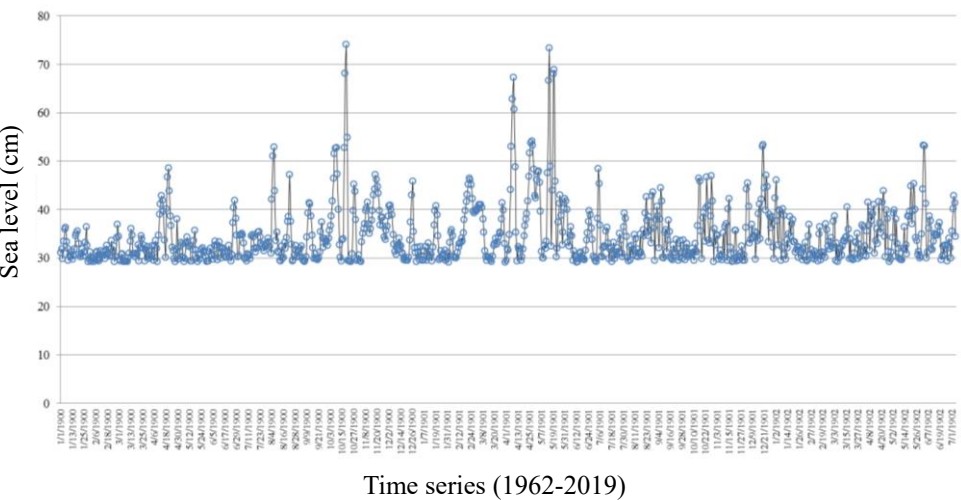

Figure 13. Surges at Busan above a threshold of 29.15cm, before clustering


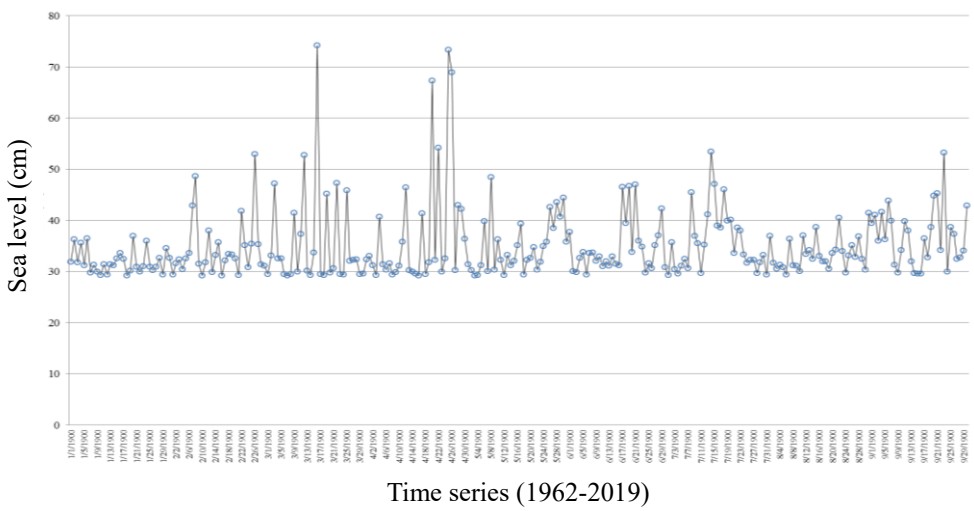

Figure 13. Surges at Busan above a threshold of 29.15 cm, after clustering

### 3.2.2 Relationship among storm-surge parameters

*Storm-surge parameters*

Storm surges are characterised by four major parameters: peak time, peak height, duration, and rise ratio. Peak time follows a gamma distribution because POT produces a Poisson process of exceedance occurrence, and the waiting times between consecutive exceedances in a Poisson process are, by definition, exponentially distributed (Lopeman et al., 2015). For peak times (interarrival times), this study therefore uses a gamma (exponential) distribution.

For peak height, on the other hand, GPD is typically used, because some representation-theorem results from extreme-value

statistics indicate that, if the cluster maxima follow a Poisson process, then the intensity – in this case, height – of the cluster peaks follows a GPD distribution (Lopeman et al., 2015; Zhong et al., 2014). However, a Weibull distribution has been applied to peak storm-surge heights in this study because it fits the data better, especially with regard to the right-hand tail.

Because the rise ratio does not appear to be evenly distributed along the interval [0, 1], a beta distribution was used because the rise ratio is by definition between 0 and 1, and such a distribution is commonly used to model continuous random variables

that occur within that range (Lopeman et al., 2015).

Duration follows a lognormal distribution, which was used for the following two reasons, previously articulated by Lopeman et al. (2015). First, it models a continuous random variable, duration, which by definition is positive. And second, it is quite flexible: i.e., having two parameters, it can fit the data better than other distributions with just one, such as exponential distribution.

*Parameter interrelationships*

Figures 15, 16, and 17 indicate the lack of any clear relationship between rise ratio, on the one hand, and either duration or exceedance, on the other. However, peak exceedance and cluster duration appear to have a linear relationship.

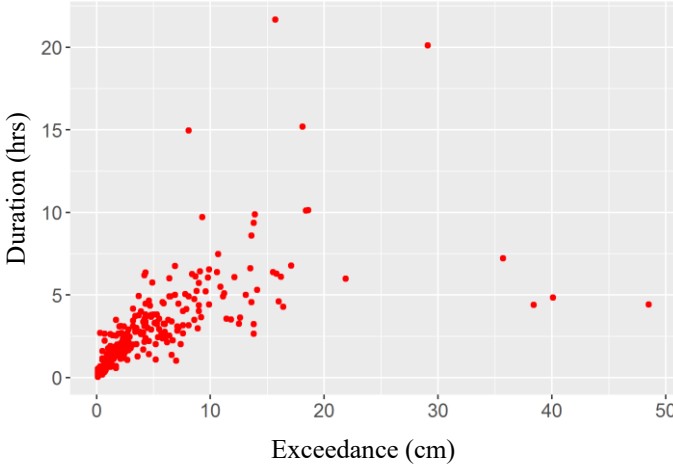

Figure 15. Relationship between exceedance and duration

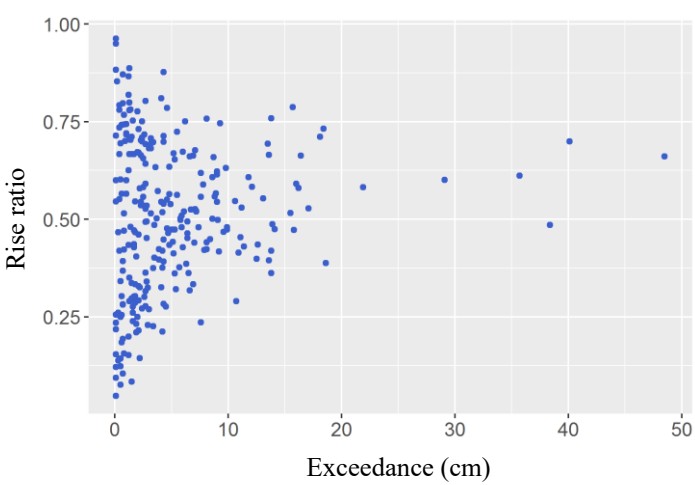


Figure 16. Relationship between exceedance and rise ratio

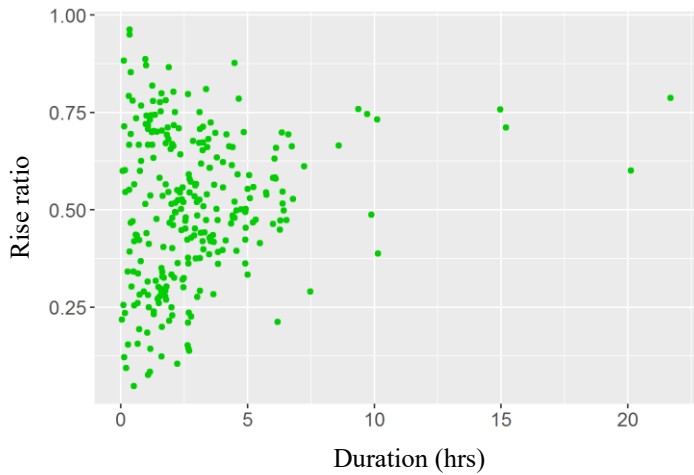

Figure 17. Relationship between duration and rise ratio

## 4 Results and analysis

### 4.1 Storm surge simulation

After finding the threshold that resulted from a given target rate, we computed interarrival times, rise ratios, peak height, and cluster duration for each exceedance cluster. These figures were then grouped by season (the year being divided for this purpose into a cold season, lasting from 1 December through 31 May, and a warm season, 1 June through 30 November), and such groups were used to estimate the parameters of the statistical model via MLE. For the reasons given in the previous section, the interarrival times for each season were fitted with an exponential distribution; the rise ratios with a beta distribution; and the peak heights with a Weibull distribution in which the location parameter was equal to the threshold. Detailed descriptions of how we applied each of these methods are provided in turn below.

*Maximum likelihood estimation*

If we assume that an independent and identically distributed data sample $(x_1, ..., x_n)$ is observed from a population with a distribution of interest parametrised by an unknown variable $\theta$, which the researcher wants to estimate, the MLE estimator $\hat{\theta}_{MLE}$ is defined as

$$\hat{\theta}_{MLE}(x_1, ..., x_n) = argmax_{\theta_0} \prod_{i=1}^{n} f(x_i; \theta_0),$$  (13)

where $f(\cdot; \theta)$ denotes the probability-density function of the distribution of interest, parametrised by $\theta_0$. The distributions of interest for the data in this study were chosen as follows:

(1) $T_i \sim Exponential(\lambda)$, where $T_i$ denotes the interarrival time between the peak of the $i - 1$th cluster and the peak of the $i$-th cluster. This distributional assumption is equivalent to assuming that a Poisson process governs peak-surge arrivals.

(2) $\Phi_i \sim Beta(\alpha, \beta)$, where $\Phi_i$ denotes the rise ratio of the $i$-th cluster.

(3) $\prod_i \sim GPD(\xi, \sigma, \theta^*)$, where $\prod_i$ denotes the peak surge height of the $i$-th cluster, and $\theta^*$ the selected threshold.

For the exponential distribution/interarrival times, the exact solutions of the maximisation problem stated above can be derived in closed form. For the GPD distribution/peak exceedances, and the beta distribution/rise ratios, the problem is solved numerically. A full description of the MLE algorithm for interarrival times, rise ratios, and peak exceedances is detailed below.

     (1) Input:

(a) Observed interarrival times $t_1, \ldots, t_C$ of the clusters' surge peaks.

         (b) Observed rise ratios $\phi_1, \ldots, \phi_C$.

         (c) Observed peak surge heights $\gamma_1, \ldots, \gamma_C$.

         (d) Number of clusters $C$.

         (e) Threshold rate $\theta^*$.

(2) Output:

        Maximum-likelihood estimates of the model parameters $\hat{\lambda}_{\text{MLE}}, \hat{\alpha}_{\text{MLE}}, \hat{\beta}_{\text{MLE}}, \hat{\xi}_{\text{MLE}},$ and $\hat{\sigma}_{\text{MLE}}$.

     (3) Procedure:

     (a) Compute $\hat{\lambda}_{MLE}$ for the exponential interarrival rate $\lambda$ as:

$$\hat{\lambda}_{MLE} = \left( \sum_{c=1}^{C} t_c \right)^{-1} \tag{14}$$

(b) Compute $\hat{\alpha}_{\text{MLE}}$ and $\hat{\beta}_{\text{MLE}}$ for the beta parameters $\alpha$ and $\beta$, by numerically solving the following first-order equations,

$$C\left(\psi(\hat{\alpha}_{MLE} + \hat{\beta}_{MLE}) - \psi(\hat{\alpha}_{MLE})\right) + \sum_{c=1}^{C} \log \phi_i = 0; \tag{15}$$

$$C\left(\psi(\hat{\alpha}_{MLE} + \hat{\beta}_{MLE}) - \psi(\hat{\alpha}_{MLE})\right) + \sum_{c=1}^{C} \log(1 - \phi_i) = 0, \tag{16}$$

in which $\psi(\cdot)$ denotes the digamma function.

     (c) Compute $\hat{\xi}_{\text{MLE}}$ and $\hat{\sigma}_{\text{MLE}}$ for the GPD parameters $\xi$ and $\sigma$. Further details on this estimation can be found in the

documentation provided with the ismev package (Heffernan and Stephenson, 2012).

     (d) Return $\hat{\lambda}_{\text{MLE}}, \hat{\alpha}_{\text{MLE}}, \hat{\beta}_{\text{MLE}}, \hat{\xi}_{\text{MLE}},$ and $\hat{\sigma}_{\text{MLE}}$ to step 3(a), above.

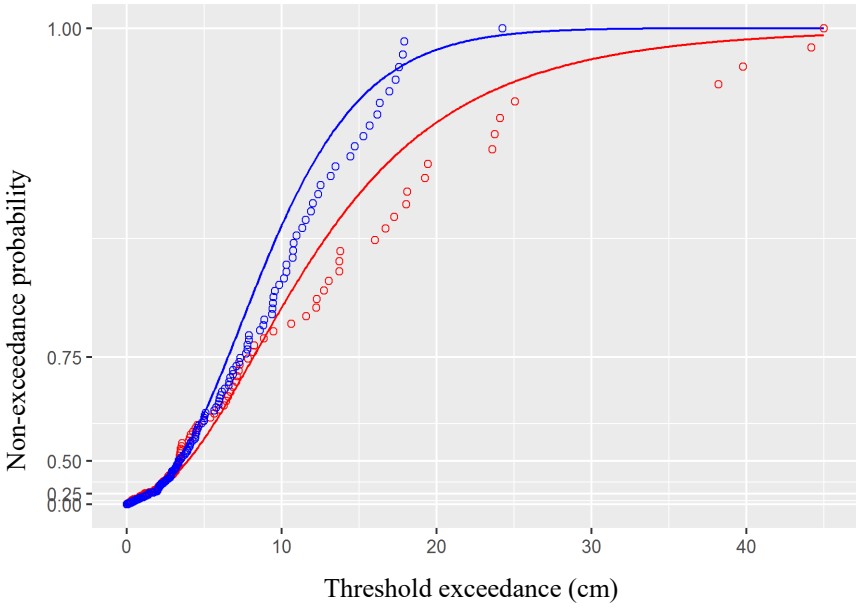

Figure 18. Non-exceedance probability plot of surge height at target rate=5.0

Based on our simulations, exceedances water height above the designated threshold were computed using MLE estimates.
Table 11 presents the distribution parameters of the storm-surge parameters that were computed, each using a different
probability model. These distribution parameters were based on the exceedance above the algorithmically designated threshold
of 29.15cm, mentioned above.

Table 11. Probability-distribution parameters of the storm-surge parameters

| Season | GPD | | Beta | |
|--------|-----|-----|------|-----|
| | $\xi$ | $\sigma$ | $\alpha$ | $\beta$ |
| Cold | 0.02 | 0.34 | 3.12 | 3.45 |
| Warm | 0.51 | 0.33 | 2.87 | 1.89 |




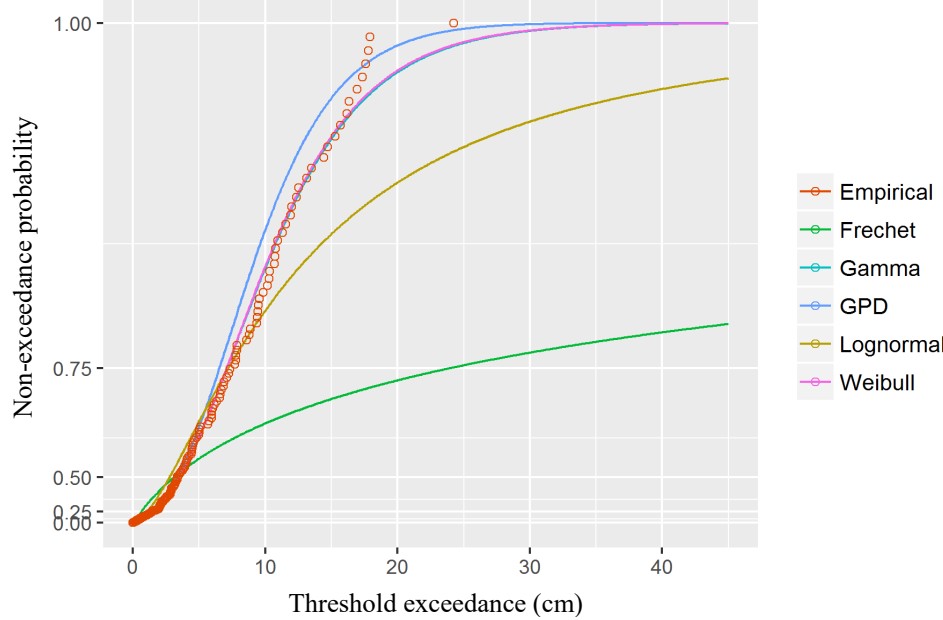

Figure 19. Fits of six types of distributions of non-exceedance probability, cold season, target rate=5.0

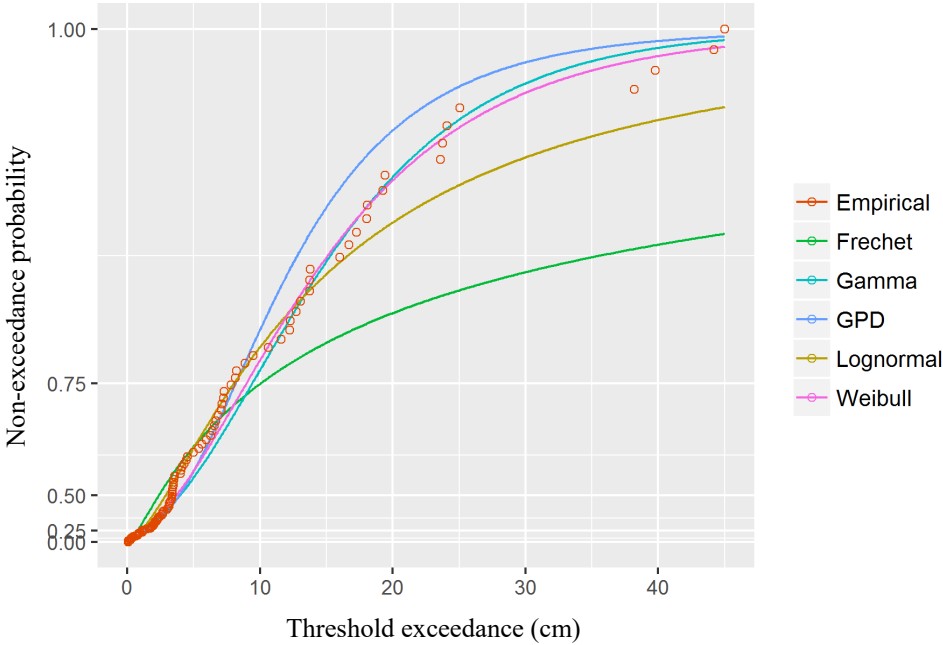

Figure 20. Fits of six types of distributions of non-exceedance probability, warm season, target rate=5.0

Figure 18 shows the GPD cumulative-distribution function as estimated by MLE, and the empirical-distribution function, with the latter shown as dots. Each dot represents the observed proportion of exceedances below a certain height in a given season (blue: cold season, red: warm season), while the corresponding value on the fitted line of the same season gives the probability that the exceedances are below that height, per the estimated GPD distribution.

We also fit our empirical data to five different probability-distribution models – i.e., Fréchet, Gamma, GPD, Lognormal, and Weibull – as seen in Figures 19 and 20, using the case of storm-surge data. Calculation of mean squared error between the probability models and the empirical data revealed that the Gamma and Weibull distributions had the best fit to the data for both cold and warm seasons when MLE was used for estimating parameters of the probability model. These findings support previous ones by Bardsley (2019) regarding the Weibull distribution's appropriateness to extreme-value estimation. According to Bardsley, such a distribution could explain enough to enable extrapolation of the degree beyond the utilised data history, provided that the scale and shape parameter of the distribution are positive (meaning that the probability model has a good fit to the data). In the case of our own research, the shape and scale parameters were 1.87 and 5.21, respectively, indicating that the Weibull distribution model will likely have a good fit to large amounts of data beyond the dataset we used.

**5 Conclusion**

Typhoons cause numerous fatalities and immense property damage, and their frequency has recently been increasing. Nevertheless, typhoon risk assessments are not yet sufficiently comprehensive to estimate either the damage levels from such events, or the probability of their occurrence. If they are to effectively plan for typhoons, governments and the insurance industry will need accurate estimates of both. Prompted by the high levels of damage inflicted by the high surge during South Korea's most severe typhoon, Maemi, this research has estimated the risk of storm surges through non-exceedance probability using MLE. Specifically, we estimated extreme storm surges' non-exceedance probability in accordance with their water levels, with such levels serving as references for non-exceedance probability above a certain threshold. We applied various methodologies to obtain more reliable thresholds, and a threshold-selection algorithm that utilised target rate and number of clusters to more accurately predict the height threshold. Additionally, we separated storm surges into cold-season and warm-season ones, as this allowed more reliable estimations, given their different frequencies in these seasons. Three parameters – exceedance, rise ratio, and duration – were separated from the storm surges and compared to ascertain their relationship. This established that exceedance and duration have a quite strong linear relationship. In previous research, total water level was utilised to estimate the possibility of future occurrences, but such an approach could lead to inaccurate results, for the reasons mentioned in the Literature Review section, above. Accordingly, in this study, we subcategorised total water levels into predicted, observed, and surge levels. Once that had been done, surge level was found to be the main factor influencing damage to coastal infrastructure, and thus, only it was applied to our estimates of non-exceedance probability.

Based on a quantitative risk assessment for extreme storm surges in a city on the Korean Peninsula that was severely damaged by Typhoon Maemi due to its geographical characteristics, this study has proposed a risk-management approach to such natural

hazards based on the non-exceedance probabilities of extreme storm surges. Various probability-distribution models were tested within this framework to explore clustering and threshold-selection methods, and Weibull distribution was found to have the best fit to our empirical data. Our results suggest that the use of various probability models, clustering, and separation of tidal-gauge data as described above could all benefit the accuracy of natural-hazard return prediction. The present study's findings also confirm non-exceedance probability as a useful, geographically sensitive tool for government agencies, insurance companies, and construction companies conducting risk assessments, setting insurance prices, preparing safety guidelines, and setting policies aimed at reducing typhoon-related damage and financial losses.

Although the present research investigated various non-exceedance probability distributions of typhoon-driven storm surges, it only used a single extreme event in a specified region. As such, its findings may not be applicable to other regions, each of which has its own unique weather conditions, geographic features, and tidal characteristics. Future research should therefore include tidal and environmental data from a range of different regions and various extreme events to test the present study's findings. Also, various natural-hazards indicators and environmental factors such as wind speed, pressure, rainfall, landslides, distance to waterways, and so forth may be useful variables in estimating the exceedance probabilities of typhoons and other natural hazards, and thus be beneficial to risk assessment and mitigation. Also, it should be borne in mind that much of the tidal-gauge data that this study utilised was from the fairly distant past. Thus, in similar future studies, efforts should be made to ensure that such data are reliable, especially in light of climate-change-driven patterns in sea-level behaviour.

Return periods based on various non-exceedance probability models should also be considered in future research, insofar as elaborated return-period estimation can be utilised to improve disaster-relief and emergency-planning efforts. Our comparison of various probability models to find the best fitting distribution models could be adapted to the simulation of time series of the past typhoons, and the collected simulated storm-surge time series then used to estimate typhoons' return periods using bootstrapping of the exceedance data. Potentially, this would provide more exact return periods with confidence intervals. Lastly, future work on return periods should take account of trends in sea-level change, driven by climate change, which already pose a non-negligible risk to coastal buildings and other infrastructure. Advanced statistical methods such as Monte Carlo simulation, as well as deep-learning techniques, could be applied to make typhoon return-period estimates even more accurate.

**Acknowledgement**

This research was funded by Hanyang University ERICA.

**Authorship contribution statement**

**Sang-Guk Yum:** Conceptualization, Methodology, Data curation, Investigation, Project administration, Resources, Supervision, Resources, Writing – original, review & editing.

**Hsi-Hsien Wei:** Data curation, Investigation, Resources, Writing – review & editing.

**Sung-Hwan Jang:** Methodology, Software, Validation, Writing - review & editing.

**Code/Data availability**

The data presented in this research are available from the corresponding author by reasonable request.

**Declaration of competing interest**

The authors declare that they have no known competing financial interests or personal relationships that could have appeared to influence the work reported in this paper.

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
