# Peer review of "Estimation of the non-exceedance probability of extreme storm surges in South Korea using tidal-gauge data"

_Natural Hazards and Earth System Sciences, 2020_

## Referee Comment (RC1) · Anonymous Referee #1 · 2 Dec 2020

The paper presents an interesting and potentially publishable piece of research that would be of interest to both the wider academic research community as well as stakeholder in the construction industry. However, I recommend a minor revision in terms of contents and editing.

1. Research framework: The main shortcoming is a lack of clear research framework that should contain a clearly set main research goal and objectives, research methodology and research hypothesis.

2. Conclusions should be more clearly explained with additional discussion and the value of findings/analysis

2. Conclusions should be more clearly explained with additional discussion and the value of findings/analysis

---

## Referee Comment (RC2) · Anonymous Referee #2 · 17 Dec 2020

Dear authors,

I read with interest your manuscript titled: "Identifying the non-exceedance probability of extreme storm surges as a component of natural-disaster management using tidal-gauge data from Typhoon Maemi in South Korea". The manuscript introduces a novel methodology for deriving non-exceedance probability diagrams of extreme surge storms. Clustered separated peaks-over-threshold simulation was developed, and various probability density function models were fitted to the empirical data for investigating the risk of storm surge height. Weibull probability density distribution was found to fit

the empirical data. This manuscript introduce a novel simulation method for derivation of exceedance diagrams of storm surges that can contribute to many other natural hazards phenomena such as floods, forest fires, etc. The paper deserves minor revisions as follows: 1. The title of the manuscript is too long consider shorter title such as: "Non-exceedance probability of extreme storm surges using tidal-gauge". 2. The abstract does not reflect the novelty of the methodology. Consider revision. 3. Please add a research framework diagram, emphasize the core phases of the methodology, particularly: the Threshold selection iterative process and the clustering of storm surge data. 4. The statistics analysis presentation: it is suggested to add Analysis of Variance output data for all regression and probabilistic distribution goodness of fit as follows: a. Please provide Coefficient of determination and correlationcofficiet, regression variance and Standard Error (SE) of the sea level fluctuations in Figure 4. Perhaps add a Table; b. Please provide detailed statistical data of the different probabilistic distributions I Figures 17-19. Please present the parameters of the selected distribution (Weibull) and discuss this with reference to the literature. c. Please discuss the statistical significance of the model (P=Value, R2, C.I. P.I.). 5. Please elaborate the literature review with reference to up-to-date publications, please refer to (Ke et al. 2018; Catalano et al. 2019; McInnes et al. 2016; Silva-González et al. 2017; Hisamatsu et al. 2020; Buchana and McSharry 2019; Chen et al. 2019; Wahl et al. 2015; Bermúdez et al. 2019; Davies et al. 2017; Fawcett et al. 2016) with regards to exceedance diagrams and to (Zhu et al. 2017; Yum et al. 2020; Ke et al. 2018) with regards to storm surge risk assessment.

6. Please see further comments and typo-edit in the attached. 7. Some of the Figures need Legend and improved resolution. 8. Some references in the text are miss from the bibliographic list. Good Luck!

Ref.

Bermúdez, M., Cea, L., and Sopelana, J. (2019). "Quantifying the role of individual flood drivers and their correlations in flooding of coastal river reaches." Stoch Environ Res Risk Assess, 33(10), 1851-1861, https://doi.org/10.1007/s00477-019-01733-8/. Buchana, P., and McSharry, P. E. (2019). "Windstorm risk assessment for offshore wind farms in the North Sea." Wind Energy (Chichester, England), 22(9), 1219-1229, https://doi.org/10.1002/we.2351/. Catalano, A. J., Broccoli, A. J., Kapnick, S. B., and Janoski, T. P. (2019). "High-Impact Extratropical Cyclones along the Northeast Coast of the United States in a Long Coupled Climate Model Simulation." Journal of Climate, 32(7), 2131-2143, https://doi.org/10.1175/JCLI-D-18-0376.1/. Chen, Y., Li, J., Pan, S., Gan, M., Pan, Y., Xie, D., and Clee, S. (2019). "Joint probability analysis of extreme wave heights and surges along China's coasts." Ocean Engineering, 177 97-107, https://doi.org/10.1016/j.oceaneng.2018.12.010/. Davies, G., Callaghan, D. P., Gravois, U., Jiang, W., Hanslow, D., Nichol, S., and Baldock, T. (2017). "Improved treatment of non-stationary conditions and uncertainties in probabilistic models of storm wave climate." Coastal Engineering (Amsterdam), 127 1-19, https://doi.org/10.1016/j.coastaleng.2017.06.005/. Fawcett, L., Fawcett, L., Walshaw, D., and Walshaw, D. (2016). "Sea-surge and wind speed extremes: optimal estimation strategies for planners and engineers." Stoch Environ Res Risk Assess, 30(2), 463-480, https://doi.org/10.1007/s00477-015-1132-3/. Hisamatsu, R., Tabeta, S., Kim, S., and Mizuno, K. (2020). "Storm surge risk assessment for the insurance system: A case study in Tokyo Bay, Japan." Ocean & Coastal Management, 189, https://doi.org/10.1016/j.ocecoaman.2020.105147/. Ke, Q., Jonkman, S. N., van Gelder, P. H. A. J. M, and Bricker, J. D. (2018). "Frequency Analysis of Storm-Surge-Induced Flooding for the Huangpu River in Shanghai, China." Journal of Marine Science and Engineering, 6(2), https://doi.org/10.3390/jmse6020070/. McInnes, K., Hoeke, R., Walsh, K., O'Grady, J., and Hubbert, G. (2016). "Application of a synthetic cyclone method for assessment of tropical cyclone storm tides in Samoa." Nat Hazards, 80(1), 425-444, https://doi.org/10.1007/s11069-015-1975-4/. Silva-González, F., Heredia-Zavoni, E., and Inda-Sarmiento, G. (2017). "Square Error Method for threshold estimation in extreme value analysis of wave heights." Ocean Engineering, 137 138-150, https://doi.org/10.1016/j.oceaneng.2017.03.028/.

Wahl, T., Mudersbach, C., and Jensen, J. (2015). "Statistical Assessment of Storm Surge Scenarios Within Integrated Risk Analyses." Coastal Engineering Journal, 57(1), https://doi.org/10.1142/s0578563415400033/. Yum, S., Kim, J. H., and Wei, H. (2020). "Development of vulnerability curves of buildings to windstorms using." Journal of Building Engineering, Article in Press. Zhu, Y., Xie, K., Ozbay, K., Zuo, F., and Yang, H. (2017). "Data-driven spatial modeling for quantifying networkwide resilience in the aftermath of hurricanes Irene and Sandy." Transp.Res.Rec., 2604(1), 9-18.

---

## Referee Comment (RC3) · Anonymous Referee #3 · 19 Dec 2020

The manuscript details an important and highly relevant topic: Development of probability of storm surge occurrences which seeks to develop a risk analysis for predicting natural hazards. The motivation and the importance of the study are clearly presented, and it has fruitful information and a review of the state-of-the art regarding empirical analysis for storm surges. The proposed minor revisions are followed: 1. The methodology paragraph is highly important in this paper. Please provide a general approach and workflow. 2. Please improve the quality of the figures. 3. Please emphasize the contribution of the research 4. Please add more relevant literature review with up-to-

date

---

## Referee Comment (RC4) · Anonymous Referee #4 · 28 Dec 2020

Abstract : Since this is a scientific paper, it is recommended to summarize the research methodology and conclusions (major outcomes) in more detail.

Line 43 : I can't fine this paper(Hwang and Deodatis 2013) in your reference. Check please.

Line 71 : I recommend that the names of Sea of Japan and East Sea should be written as "East/Japan Sea" together.

Line 85 : In your Table 2 & 3, Fill the last column in line1 -> Total Sum & Average in

line2∼3 -> Incidence(sum) & Incidence(average)

Line 94∼96 : South Korea operated 17 tidal gauge stations. -> South Korea operated 46 tidal gauge stations. And (n=5), (n=10), (n=2)∼ Check please.

Line 99 : These expressions are already expressed in Figure 1 and are redundant, so please remove them.

Line 105 : Authors did not to describe about QC processes(i.e. interpolation, outliers etc) of tidal gauge data, cause tidal stations have lots of errors in raw data. Rising rate of MSL can be varied considerably according to the OC processes.

Line 106 : Highest recorded water levels -> Recorded highest water levels

Line 110 : Table 5 is presented but not cited in the text.

Line 115 : In Table 5, New Busan -> New Busan Port

Line 129 : Figure 4 and 5 -> Figure 3 and 4

Line 130 : I think it would be appropriate to delete Figure 3 because it is not important in the context described in the paper.

Line 132 : In Figure 3 & 4, Sea level (cm) -> Mean sea level (cm)

Line 136 : Figure 4. Sea-level fluctuations around the mean, Busan -> Figure 4. Mean Sea-level fluctuations in Busan

Line 139 : "Looking at the sea-level history in Figure 3, it is clear that the data trend between 1956 and 1961 is anomalous. As this may have been due to quality-control issues with the observations from that period, it has been excluded from this study, and only data from 1962 to 2019 have been used, as shown in Figure 4." -> If you accept the recommendation, please delete this context.

Line 154 : In Table 7, New Busan -> New Busan Port

Line 155 : As can be seen from Table 7,

Line 185 : I can't fine this paper(Lee et al. 2008) in your reference. Check please.

Line 201 : description on Previous research results

Line 231 : Introduction on return period of Hurricane using GEV, MLE, CSPS

Line 268 : I can't fine this paper(Pickands, 1975 & Scarrott and Macdonald 2012) in your reference. Check please.

Line 333 : Description on the surge height caused by typhoon Maemi.

Line 339 : I can't find where is the Figure 7 in the text.

Although the return period is importantly mentioned at the beginning, it is not presented in the results, and that only mentioning that further research is needed may lower the quality and justification of the paper, so additional supplementation is required. Since this is a scientific paper, the conclusion should be explained in detail.

I look forward to seeing you again with the revised thesis.

Happy New year!

———————————————————

---

## Author Comment (AC1) · 18 May 2021

My co-authors and I would like to express our gratitude to the reviewers for their constructive feedback and suggestions for strengthening our research. The changes we have made to the attached file in response to such feedback and suggestions have been highlighted in blue to facilitate their identification. I would also like to offer my apologies for the length of time it took us to prepare this response.

**Referee #1**

The paper presents an interesting and potentially publishable piece of research that would be of interest to both the wider academic research community as well as stakeholder in the construction industry. However, I recommend a minor revision in terms of contents and editing.

1. Research framework: The main shortcoming is a lack of clear research framework that should contain a clearly set main research goal and objectives, research methodology and research hypothesis.
- We are grateful for this insightful suggestion. Our general approach and workflow regarding estimating non-exceedance probability of extreme surges using tidal gauge data. These can be seen below.

[revised manuscript text omitted]

---

## Author Comment (AC2) · 18 May 2021

My co-authors and I would like to express our gratitude to the reviewers for their constructive feedback and suggestions for strengthening our research. The changes we have made to the attached file in response to such feedback and suggestions have been highlighted in blue to facilitate their identification. I would also like to offer my apologies for the length of time it took us to prepare this response.

**Referee #2**

I read with interest your manuscript titled: "Identifying the non-exceedance probability of extreme storm surges as a component of natural-disaster management using tidal-gauge data from Typhoon Maemi in South Korea". The manuscript introduces a novel methodology for deriving non-exceedance probability diagrams of extreme surge storms. Clustered separated peaks-over-threshold simulation was developed, and various probability density function models were fitted to the empirical data for investigating the risk of storm surge height. Weibull probability density distribution was found to fit the empirical data. This manuscript introduces a novel simulation method for derivation of exceedance diagrams of storm surges that can contribute to many other natural hazards phenomena such as floods, forest fires, etc. The paper deserves minor revisions as follows:

1. The title of the manuscript is too long consider shorter title such as: "Nonexceedance probability of extreme storm surges using tidal-gauge".
-    We appreciate this insightful comment, and as recommended, have modified the title of the manuscript to better reflect its content.

Estimation of the non-exceedance probability of extreme storm surges in South Korea using tidal-gauge data

2. The abstract does not reflect the novelty of the methodology. Consider revision.
-    We are very grateful to the reviewer for providing this important advice. Our revisions in response to the above comments can be found in the revised Abstract. It can also be seen below.

Global warming, one of the most serious aspects of climate change, can be expected to cause rising sea levels. These, in turn, have been linked to unprecedentedly large typhoons that can cause flooding of low-lying land, coastal invasion, seawater flows into rivers and groundwater, rising river levels, and aberrant tides. To prevent typhoon-related loss of life and property damage, it is crucial to accurately estimate storm-surge risk. This study therefore develops a statistical model for estimating such surges' probability, based on surge data pertaining to Typhoon Maemi, which struck South Korea in 2003. Specifically, estimation of non-exceedance probability models of the

typhoon-related storm surge was achieved via clustered separated peaks-over-threshold simulation, while various distribution models were fitted to the empirical data for investigating the risk of storm surges reaching particular heights. To explore the non-exceedance probability of extreme storm surges caused by typhoons, a threshold algorithm with clustering methodology was applied. To enhance the accuracy of such non-exceedance probability, the surge data was separated into three different components: predicted water level, observed water level, and surge. Sea-level data from when Typhoon Maemi struck was collected from a tidal gauge station in the City of Busan, which is vulnerable to typhoon-related disasters due to its geographical characteristics. Fréchet, Gamma, log-normal, Generalised Pareto, and Weibull distributions were fitted to the empirical surge data, and the researchers compared each one's performance at explaining the non-exceedance probability. This established that Weibull distribution was better than any of the other distributions for modeling Typhoon Maemi's peak total water level. Although this research was limited to one city in the Korean Peninsula and one extreme weather event, its approach could be used to reliably estimate non-exceedance probabilities in other regions where tidal gauge data are available. In practical terms, the findings of this study, and future ones adopting its methodology, will provide a useful reference for designers of coastal infrastructure.

3. Please add a research framework diagram, emphasize the core phases of the methodology, particularly: the Threshold selection iterative process and the clustering of storm surge data.

- We are grateful for this insightful suggestion. Our general approach, workflow, threshold selection interactive process, and clustering regarding estimating non-exceedance probability of extreme surges using tidal gauge data. These can be seen below.

[Figure]

Figure 4. General approach and workflow

[Figure]

Figure 11. Clustering flowchart

[Figure]

Figure 6. Threshold-selection flowchart

4. The statistics analysis presentation: it is suggested to add Analysis of Variance output data for all regression and probabilistic distribution goodness of fit as follows:
   1) Please provide Coefficient of determination and correlation coefficient, regression variance and Standard Error (SE) of the sea level fluctuations in Figure 4. Perhaps add a Table
   2) Please provide detailed statistical data of the different probabilistic distributions I Figures 17-19.
   3) Please present the parameters of the selected distribution (Weibull) and discuss this with reference to the literature.
   4) Please discuss the statistical significance of the model (P=Value, R2, C.I. P.I.).
- Thank you for your comment. Pursuant to this reviewer comment, the deeper description of statistical analysis was discussed. These can be seen below.

KHOA makes hourly observations of water height at the Busan tidal-gauge station, and

the annual means presented in this paper have been calculated from that hourly data. As can be seen in Figure 3, plotting MSL for each year confirms that short-term water-level variation merely masks the long-term trend of sea-level increase. Therefore, on the assumption that MSL variation was a function of time, a linear regression was performed, with the resulting coefficient of slope indicating the rate of increase (Yoon and Kim, 2012). The data utilised to estimate MSL for the tidal gauge station in Busan was provided by KHOA, which performed quality control on it before releasing it to us. Additionally, however, a normality test was performed, and the results (as shown in Table 7) indicated that the hourly sea-level data followed a normal distribution, at a significance >0.05. The Kolmogorov-Smirnov normality test was adopted as being well-suited to datasets containing more than 30 items.

Table 7. Kolmogorov-Smirnov normality test of sea-level fluctuation data from Busan tidal-gauge station

|  | Statistic | df | Significance | Pearson correlation |
| --- | --- | --- | --- | --- |
| Sea level fluctuation | 0.084 | 473352 | 0.200 | 0.96 |

As can be seen in Figure 3, the average rate of increase in MSL at Busan's tidal-gauge station from 1962 to 2019 was 2.4mm per year, yielding a difference of 16.31cm between the end of that period and the beginning. This finding is broadly in line with Yoon and Kim's (2012), that the rate of MSL increase around the Korean Peninsula as a whole between 1960 and 2010 was about 2.9mm/year. Also, linear regression analysis of the sea-level fluctuation data for 1965-2019 was utilised to discern the MSL trend. The significance level of 0.000 (<0.05) obtained via analysis of variance (ANOVA; Table 8) indicates that the regression model of sea-level fluctuations was significant. Also, its correlation coefficient (0.96) indicated a strong positive relationship between sea-level rise and recentness. The coefficient of determination ($R^2$) was utilised to describe how well the model explained the collected data. The closer $R^2$ is to 1, the better the model can predict the linear trend; and here, it was 0.74, as shown in Table 9. This means that the linear-regression model explained 74% of the sea-level variation. While this result suggests that the linear-regression analysis for sea-level fluctuation at the tidal gauge station in Busan is reliable, however, such results may not be generalisable because variation in the data could have been due to several factors, including geological variation and modification of gauge points.

Table 8. Linear-regression coefficients, sea-level fluctuations at Busan tidal-gauge

| | Non-standardised Coefficients | | Standardised Coefficients | t | Significance Probability (P-value) |
|---|---|---|---|---|---|
| | B | Standard Error | Beta | | |
| (Constant) | -422.23 | 35.022 | 0.887 | -12.06 | 0.00 |
| Sea level fluctuations | 0.246 | 0.018 | | 13.97 | 0.00 |

Table 9. Summary of analysis of variance results, sea-level fluctuations at Busan tidal-gauge station

| Model | Sum of Squares | df | Mean squares | F | Sig. | Adjusted R2 |
|---|---|---|---|---|---|---|
| Regression | 830446354.04 | 41 | 20254789.12 | 32109.38 | .000 b | 0.74 |
| Residual | 298566787.86 | 473310 | 630.81 | | | |
| Total | 1129013141.90 | 473351 | | | | |

Based on our simulations, exceedances water height above the designated threshold were computed using MLE estimates. Table 11 presents the distribution parameters of the storm-surge parameters that were computed, each using a different probability model. These distribution parameters were based on the exceedance above the algorithmically designated threshold of 29.15cm, mentioned above.

Table 11. Probability-distribution parameters of the storm-surge parameters

| | GPD | | Beta | |
|---|---|---|---|---|
| Season | $\xi$ | $\sigma$ | $\alpha$ | $\beta$ |
| Cold | 0.02 | 0.34 | 3.12 | 3.45 |

| | | | | |
|---|---|---|---|---|
| Warm | 0.51 | 0.33 | 2.87 | 1.89 |

We also fit our empirical data to five different probability-distribution models – i.e., Fréchet, Gamma, GPD, Lognormal, and Weibull – as seen in Figures 19 and 20, using the case of storm-surge data. Calculation of mean squared error between the probability models and the empirical data revealed that the Gamma and Weibull distributions had the best fit to the data for both cold and warm seasons when MLE was used for estimating parameters of the probability model. These findings support previous ones by Bardsley (2019) regarding the Weibull distribution's appropriateness to extreme-value estimation. According to Bardsley, such a distribution could explain enough to enable extrapolation of the degree beyond the utilised data history, provided that the scale and shape parameter of the distribution are positive (meaning that the probability model has a good fit to the data). In the case of our own research, the shape and scale parameters were 1.87 and 5.21, respectively, indicating that the Weibull distribution model will likely have a good fit to large amounts of data beyond the dataset we used.

5. Please elaborate the literature review with reference to up-to-date publications, please refer to;

1) with regards to exceedance diagrams

[1] Bermúdez, M., Cea, L., and Sopelana, J. (2019). "Quantifying the role of individual flood drivers and their correlations in flooding of coastal river reaches." Stoch Environ Res Risk Assess, 33(10), 1851-1861, https://doi.org/10.1007/s00477-019-01733-8/.

[2] Buchana, P., and McSharry, P. E. (2019). "Windstorm risk assessment for offshore wind farms in the North Sea." Wind Energy (Chichester, England), 22(9), 1219-1229, https://doi.org/10.1002/we.2351/.

[3] Catalano, A. J., Broccoli, A. J., Kapnick, S. B., and Janoski, T. P. (2019). "High-Impact Extratropical Cyclones along the Northeast Coast of the United States in a Long Coupled Climate Model Simulation." Journal of Climate, 32(7), 2131-2143, https://doi.org/10.1175/JCLI-D-18-0376.1/.

[4] Chen, Y., Li, J., Pan, S., Gan, M., Pan, Y., Xie, D., and Clee, S. (2019). "Joint probability analysis of extreme wave heights and surges along China's coasts." Ocean Engineering, 177 97-107, https://doi.org/10.1016/j.oceaneng.2018.12.010/.

[5] Davies, G., Callaghan, D. P., Gravois, U., Jiang, W., Hanslow, D., Nichol, S., and Baldock, T. (2017). "Improved treatment of non-stationary conditions and uncertainties in probabilistic models of storm wave climate." Coastal Engineering (Amsterdam), 127 1-19, https://doi.org/10.1016/j.coastaleng.2017.06.005/.

[6] Fawcett, L. and Walshaw, D. (2016). "Sea-surge and wind speed extremes: optimal estimation strategies for planners and engineers." Stoch Environ Res Risk Assess, 30(2), 463-480, https://doi.org/10.1007/s00477-015-1132-3/.

[7] Hisamatsu, R., Tabeta, S., Kim, S., and Mizuno, K. (2020). "Storm surge risk assessment for the insurance system: A case study in Tokyo Bay, Japan." Ocean &

Coastal Management, 189, https://doi.org/10.1016/j.ocecoaman.2020.105147/.

[8] Ke, Q., Jonkman, S. N., van Gelder, P. H. A. J. M, and Bricker, J. D. (2018). "Frequency Analysis of Storm-Surge-Induced Flooding for the Huangpu River in Shanghai, China." Journal of Marine Science and Engineering, 6(2), https://doi.org/10.3390/jmse6020070/.

[9] McInnes, K., Hoeke, R., Walsh, K., O'Grady, J., and Hubbert, G. (2016). "Application of a synthetic cyclone method for assessment of tropical cyclone storm tides in Samoa." Nat Hazards, 80(1), 425-444, https://doi.org/10.1007/s11069-015-1975-4/.

[10] Silva-González, F., Heredia-Zavoni, E., and Inda-Sarmiento, G. (2017). "Square Error Method for threshold estimation in extreme value analysis of wave heights." Ocean Engineering, 137 138-150, https://doi.org/10.1016/j.oceaneng.2017.03.028/.

[11] Wahl, T., Mudersbach, C., and Jensen, J. (2015). "Statistical Assessment of Storm Surge Scenarios Within Integrated Risk Analyses." Coastal Engineering Journal, 57(1), https://doi.org/10.1142/s0578563415400033/.

2) with regards to storm surge risk assessment;

[1] Zhu, Y., Xie, K., Ozbay, K., Zuo, F., and Yang, H. (2017). "Data-driven spatial modeling for quantifying networkwide resilience in the aftermath of hurricanes Irene and Sandy." Transp.Res.Rec., 2604(1), 9-18.

[2] Yum, S., Kim, J. H., and Wei, H. (2020). "Development of vulnerability curves of buildings to windstorms using." Journal of Building Engineering, Article in Press.

[3] Ke, Q., Jonkman, S. N., van Gelder, P. H. A. J. M, and Bricker, J. D. (2018). "Frequency Analysis of Storm-Surge-Induced Flooding for the Huangpu River in Shanghai, China." Journal of Marine Science and Engineering, 6(2), https://doi.org/10.3390/jmse6020070/.

- We are very grateful to the reviewer for providing these valuable references. As requested, we have added reviews of the studies recommended above by the reviewer. These can also be seen below.

[revised manuscript text omitted]

6. Please see further comments and typo-edit in the attached.
-    We would again like to thank Reviewer #1 for the above insightful and constructive comments on our manuscript. We hope that all of them have now been addressed, but of course, are happy to make further changes if needed.

7. Some of the Figures need Legend and improved resolution.

-    We are grateful for these constructive comments. The original Figures 1 and 2 have been revised accordingly.

[Figure]

Figure 1. Track and wind speed of Maemi, 2003

[Figure]

Figure 2. Locations of the 15 tidal-gauge stations on the western and southern coasts of South Korea as of 2003

8. Some references in the text are miss from the bibliographic list.

-    Thank you for your comment. The missing references have been added to the References list.

-

---

## Author Comment (AC3) · 18 May 2021

My co-authors and I would like to express our gratitude to the reviewers for their constructive feedback and suggestions for strengthening our research. The changes we have made to the attached file in response to such feedback and suggestions have been highlighted in blue to facilitate their identification. I would also like to offer my apologies for the length of time it took us to prepare this response.

**Referee #3**

The manuscript details an important and highly relevant topic: Development of probability of storm surge occurrences which seeks to develop a risk analysis for predicting natural hazards. The motivation and the importance of the study are clearly presented, and it has fruitful information and a review of the state-of-the art regarding empirical analysis for storm surges. The proposed minor revisions are followed:

1. The methodology paragraph is highly important in this paper. Please provide a general approach and workflow.

- We are grateful for this insightful suggestion. Our general approach and workflow regarding estimating non-exceedance probability of extreme surges using tidal gauge data. These can be seen below.

[Figure]

Figure 4. General approach and workflow

[Figure]

```
┌─────────────────────────────────────────────────────────┐
│  ┌───────────────────────────────────────────────────┐  │
│  │  Determination of target rate according to threshold │  │
│  └───────────────────────────────────────────────────┘  │
│                          │                               │
│                          ▼                               │
│  ┌───────────────────────────────────────────────────┐  │
│  │   Removal of the surge data above the threshold    │  │
│  └───────────────────────────────────────────────────┘  │
│                          │                               │
│                          ▼                               │
│  ┌───────────────────────────────────────────────────┐  │
│  │  Clustering of the surge data according to their    │  │
│  │       start and end time on the time series         │  │
│  └───────────────────────────────────────────────────┘  │
│                          │                               │
│                          ▼                               │
│  ┌───────────────────────────────────────────────────┐  │
│  │           Selection of a single maximum            │  │
│  └───────────────────────────────────────────────────┘  │
└─────────────────────────────────────────────────────────┘
```

Figure 11. Clustering flowchart

2. Please improve the quality of the figures.

- We are grateful for these constructive comments. The original Figures 1 and 2 have been revised accordingly.

[Figure]

Figure 1. Track and wind speed of Maemi, 2003

[Figure]

Figure 2. Locations of the 15 tidal-gauge stations on the western and southern coasts of South Korea as of 2003

3. Please emphasize the contribution of the research.
-    We thank you for this comment. We are grateful to the reviewer for this helpful suggestion. The following passage has accordingly been added

[revised manuscript text omitted]

4. Please add more relevant literature review with up-to- date.

- We thank you for this comment. As recommended by the reviewer, we have added reviews of the studies recommended above by the reviewer. These can also be seen below.

[revised manuscript text omitted]

---

## Author Comment (AC4) · 18 May 2021

My co-authors and I would like to express our gratitude to the reviewers for their constructive feedback and suggestions for strengthening our research. The changes we have made to the attached file in response to such feedback and suggestions have been highlighted in blue to facilitate their identification. I would also like to offer my apologies for the length of time it took us to prepare this response.

**Referee #4**

1. Abstract: Since this is a scientific paper, it is recommended to summarize the research methodology and conclusions (major outcomes) in more detail.
-    We are very grateful to the reviewer for this important advice. Our revisions in response to the above comments can be found in the revised Abstract, which can also be seen below.

Global warming, one of the most serious aspects of climate change, can be expected to cause rising sea levels. These, in turn, have been linked to unprecedentedly large typhoons that can cause flooding of low-lying land, coastal invasion, seawater flows into rivers and groundwater, rising river levels, and aberrant tides. To prevent typhoon-related loss of life and property damage, it is crucial to accurately estimate storm-surge risk. This study therefore develops a statistical model for estimating such surges' probability, based on surge data pertaining to Typhoon Maemi, which struck South Korea in 2003. Specifically, estimation of non-exceedance probability models of the typhoon-related storm surge was achieved via clustered separated peaks-over-threshold simulation, while various distribution models were fitted to the empirical data for investigating the risk of storm surges reaching particular heights. To explore the non-exceedance probability of extreme storm surges caused by typhoons, a threshold algorithm with clustering methodology was applied. To enhance the accuracy of such non-exceedance probability, the surge data was separated into three different components: predicted water level, observed water level, and surge. Sea-level data from when Typhoon Maemi struck was collected from a tidal gauge station in the City of Busan, which is vulnerable to typhoon-related disasters due to its geographical characteristics. Fréchet, Gamma, log-normal, Generalised Pareto, and Weibull distributions were fitted to the empirical surge data, and the researchers compared each one's performance at explaining the non-exceedance probability. This established that Weibull distribution was better than any of the other distributions for modeling Typhoon Maemi's peak total water level. Although this research was limited to one city in the Korean Peninsula and one extreme weather event, its approach could be used to reliably estimate non-exceedance probabilities in other regions where tidal gauge data are available. In practical terms, the findings of this study, and future ones adopting its methodology, will provide a useful reference for designers of coastal infrastructure.

2. Line 43: I can't find this paper (Hwang and Deodatis 2013) in your reference. Check please.

- We are grateful to the reviewer for pointing out that this was missing. We have now added it to the References list.

3. Line 71: I recommend that the names of Sea of Japan and East Sea should be written as "East/Japan Sea" together.

- We are grateful to the reviewer for this helpful suggestion, which has been adopted.

4. Line 85: In your Table 2 & 3, Fill the last column in line1 -> Total Sum & Average in line2-3 -> Incidence(sum) & Incidence(average)

- We thank the reviewer for this comment. The manuscript has been revised accordingly, as shown below.

Table 2. Incidence of typhoons and typhoon landfall in South Korea, 1952-2019, by month

|  | Jan. | Feb. | Mar. | Apr. | May | Jun. | Jul. | Aug. | Sep. | Oct. | Nov. | Dec. | Total |
|---|---|---|---|---|---|---|---|---|---|---|---|---|---|
| Typhoons, $n$ | 29 | 15 | 15 | 45 | 67 | 115 | 245 | 351 | 322 | 238 | 152 | 73 | 1,678 |
| Typhoons, avg. | 0.54 | 0.28 | 0.46 | 0.83 | 1.24 | 2.13 | 4.54 | 6.52 | 5.96 | 4.41 | 2.81 | 1.35 | 31.07 |
| Incidence (sum) | 0 | 0 | 0 | 0 | 1 | 18 | 65 | 70 | 45 | 5 | 0 | 0 | 206 |
| Incidence (average) | 0.0 | 0.0 | 0.0 | 0.0 | 0.02 | 0.33 | 1.2 | 1.3 | 0.87 | 0.09 | 0.0 | 0.0 | 3.81 |

Table 3. Incidence of typhoons and typhoon landfall in South Korea, 2010-19, by month

|  | Jan. | Feb. | Mar. | Apr. | May | Jun. | Jul. | Aug. | Sep. | Oct. | Nov. | Dec. | Total |
|---|---|---|---|---|---|---|---|---|---|---|---|---|---|
| Typhoons, $n$ | 4 | 3 | 4 | 5 | 12 | 18 | 33 | 43 | 56 | 34 | 16 | 7 | 235 |
| Typhoons, avg. | 0.4 | 0.3 | 0.4 | 0.5 | 1.2 | 1.8 | 3.3 | 4.3 | 5.6 | 3.4 | 1.6 | 0.7 | 23.5 |
| Incidence (sum) | 0 | 0 | 0 | 0 | 0 | 0 | 3 | 11 | 7 | 5 | 2 | 0 | 28 |
| Incidence (average) | 0 | 0 | 0 | 0 | 0 | 0 | 0.3 | 1.1 | 0.7 | 0.5 | 0.2 | 0 | 2.8 |

5. Line 94-96: South Korea operated 17 tidal gauge stations. -> South Korea operated 46 tidal gauge stations. And (n=5), (n=10), (n=2) ~. Check please.

- Pursuant to this reviewer comment, the following supplementary sentences were added.

When Typhoon Maemi struck the Korean Peninsula in 2003, South Korea was operating 17 tidal-gauge stations, of which eight had been collecting data for 30 years

or more. They were located on the western (n=5), southern (n=10), and eastern coasts (n=2).

6. Line 99: These expressions are already expressed in Figure 1 and are redundant, so please remove them.
- As recommended, the redundant expressions were removed.

7. Line 105: Authors did not to describe about QC processes (i.e. interpolation, outliers etc) of tidal gauge data, cause tidal stations have lots of errors in raw data. Rising rate of MSL can be varied considerably according to the OC processes.
- We are grateful to the reviewer for pointing this out. Accordingly, a deeper description of statistical analysis for QC processes has been added, and can also be seen below.

KHOA makes hourly observations of water height at the Busan tidal-gauge station, and the annual means presented in this paper have been calculated from that hourly data. As can be seen in Figure 3, plotting MSL for each year confirms that short-term water-level variation merely masks the long-term trend of sea-level increase. Therefore, on the assumption that MSL variation was a function of time, a linear regression was performed, with the resulting coefficient of slope indicating the rate of increase (Yoon and Kim, 2012). The data utilised to estimate MSL for the tidal gauge station in Busan was provided by KHOA, which performed quality control on it before releasing it to us. Additionally, however, a normality test was performed, and the results (as shown in Table 7) indicated that the hourly sea-level data followed a normal distribution, at a significance >0.05. The Kolmogorov-Smirnov normality test was adopted as being well-suited to datasets containing more than 30 items.

Table 7. Kolmogorov-Smirnov normality test of sea-level fluctuation data from Busan tidal-gauge station

|  | Statistic | df | Significance | Pearson correlation |
|---|---|---|---|---|
| Sea level fluctuation | 0.084 | 473352 | 0.200 | 0.96 |

As can be seen in Figure 3, the average rate of increase in MSL at Busan's tidal-gauge station from 1962 to 2019 was 2.4mm per year, yielding a difference of 16.31cm between the end of that period and the beginning. This finding is broadly in line with Yoon and Kim's (2012), that the rate of MSL increase around the Korean Peninsula as a whole between 1960 and 2010 was about 2.9mm/year. Also, linear regression analysis of the sea-level fluctuation data for 1965-2019 was utilised to discern the MSL trend. The significance level of 0.000 (<0.05) obtained via analysis of variance

(ANOVA; Table 8) indicates that the regression model of sea-level fluctuations was significant. Also, its correlation coefficient (0.96) indicated a strong positive relationship between sea-level rise and recentness. The coefficient of determination ($R^2$) was utilised to describe how well the model explained the collected data. The closer $R^2$ is to 1, the better the model can predict the linear trend; and here, it was 0.74, as shown in Table 9. This means that the linear-regression model explained 74% of the sea-level variation. While this result suggests that the linear-regression analysis for sea-level fluctuation at the tidal gauge station in Busan is reliable, however, such results may not be generalisable because variation in the data could have been due to several factors, including geological variation and modification of gauge points.

Table 8. Linear-regression coefficients, sea-level fluctuations at Busan tidal-gauge station

| | Non-standardised Coefficients | | Standardised Coefficients | t | Significance Probability (P-value) |
|---|---|---|---|---|---|
| | B | Standard Error | Beta | | |
| (Constant) | -422.23 | 35.022 | 0.887 | -12.06 | 0.00 |
| Sea level fluctuations | 0.246 | 0.018 | | 13.97 | 0.00 |

Table 9. Summary of analysis of variance results, sea-level fluctuations at Busan tidal-gauge station

| Model | Sum of Squares | df | Mean squares | F | Sig. | Adjusted R2 |
|---|---|---|---|---|---|---|
| Regression | 830446354.04 | 41 | 20254789.12 | 32109.38 | .000b | 0.74 |
| Residual | 298566787.86 | 473310 | 630.81 | | | |
| Total | 1129013141.90 | 473351 | | | | |

8. Line 106: Highest recorded water levels -> Recorded highest water levels
- We are grateful to the reviewer for pointing out this anomaly. The usual word order, "highest recorded", is now used consistently.

9. Line 110: Table 5 is presented but not cited in the text.

Line 115: In Table 5, New Busan -> New Busan Port

-    Pursuant to this comment, Table 5 is now mentioned in the main body of the manuscript, as follows.

The same approach was applied to the data from the 10 tidal-gauge stations on the south coast, as shown in Table 5 and Table 6, below.

10. Line 129: Figure 4 and 5 -> Figure 3 and 4
-    As recommended, the figure numbers have been corrected.

11. Line 130: I think it would be appropriate to delete Figure 3 because it is not important in the context described in the paper.
-    As recommended, this figure has been removed.

12. Line 132: In Figure 3 & 4, Sea level (cm) -> Mean sea level (cm)

Line 136: Figure 4. Sea-level fluctuations around the mean, Busan -> Figure 4. Mean Sea-level fluctuations in Busan

-    Pursuant to this reviewer comment, the wording of these captions has been standardized to "Mean Sea-level fluctuations".

13. Line 139: "Looking at the sea-level history in Figure 3, it is clear that the data trend between 1956 and 1961 is anomalous. As this may have been due to quality-control issues with the observations from that period, it has been excluded from this study, and only data from 1962 to 2019 have been used, as shown in Figure 4." -> If you accept the recommendation, please delete this context.
-    We are grateful for these constructive comments. As recommended, the 1956-61 data have been removed from analysis, and the text modified accordingly.

14. Line 154: In Table 7, New Busan -> New Busan Port
-    We thank for your comment. The term "Port of New Busan" is now used consistently throughout.

15. Line 155: As can be seen from Table 7
-    We thank the reviewer for pointing this out. The table number has now been corrected.

16. Line 185: I can't find this paper (Lee et al. 2008) in your reference. Check please.
-    We are grateful to the reviewer for pointing this out. The missing reference has now been added to the References list.

17. Line 201: description on Previous research results

Line 231: Introduction on return period of Hurricane using GEV, MLE, CSPS

- We are grateful for this insightful suggestion. As requested, we have added reviews of the relevant studies, as shown below.

[revised manuscript text omitted]

18. Line 268: I can't find this paper (Pickands, 1975 & Scarrott and Macdonald 2012) in your reference. Check please.

- We are grateful to the reviewer for noticing this. The missing references have been added to the References list.

19. Line 333: Description on the surge height caused by typhoon Maemi.

- As recommended, we have added more description, which can be seen below.

The Korean Peninsula is bounded by three distinct sea-systems, generally known in English as the Yellow Sea, the Korea Strait, and the East Sea / Sea of Japan. This characteristic has often led to severe damage to its coastal regions. According to the Korea Ocean Observing and Forecasting System (KOOFS), Typhoon Maemi in September 2003 had a maximum wind speed of 54 metres per second (m/s), and these strong gusts caused an unexpected storm surge. This event caused US$3.5 billion in property damage, as shown in Table 1. All three of the highest peaks ever recorded by South Korea's tidal-gauge stations also occurred in that month.

Table 1. Largest typhoons to have struck the Korean Peninsula

| Name | Date | Amount of Damage (US$) | Max. Wind Speed (10 min. avg., m/s) |
|------|------|------------------------|-------------------------------------|
|      |      |                        |                                     |

| | | | |
|---|---|---|---|
| Rusa | 30 Aug.-1 Sep. 2002 | 4.3 billion (1st) | 41 |
| Maemi | 12-13 Sep. 2003 | 3.5 billion (2nd) | 54 |
| Bolaven | 25-30 Aug. 2012 | 0.9 billion (3rd) | 53 |

The most typhoon-heavy month in South Korea is August, followed by July and September, with two-thirds of all typhoons occurring in July and August. Tables 2 and 3, below, present statistics about typhoons in South Korea over periods of 68 years and 10 years ending in 2019, respectively; and Figure 1 shows the track of Typhoon Maemi from 4-16 September 2003. As can be seen from Figure 1, Typhoon Maemi passed into Busan from the southeast, causing direct damage upon landfall, after which its maximum 10-minute sustained wind speed was 54 m/s. Typhoon Maemi prompted the insurance industry, the South Korean government, and many academic researchers to recognise the importance of advance planning and preparations for such storms, as well as for other types of natural disasters.

Because observed sea level usually differs from predicted sea level, Figure 5 depicts the former (as calculated through harmonic analysis) in blue. Predicted sea levels are shown in green, and surge height in red. As the figure indicates, the highest overall water level coincided with the highest surge during Typhoon Maemi, i.e., at 21:00 on 12 September 2003. Given a total water height of 211cm, the surge height was calculated as 73.35cm. The unexpectedly large height of the surge induced by Typhoon Maemi caused US$3.5 billion in property damage and many causalities in Busan, as mentioned in Table 1.

[Figure]

Figure 5. Observed (green), predicted (blue), and residual (red) water levels at Busan during Typhoon Maemi

20. Line 339: I can't find where is the Figure 7 in the text.

- The original Figure 7 ("Threshold-selection flowchart") has now been renumbered as Figure 6, both in its caption and in references to it in the main text.

21. Although the return period is importantly mentioned at the beginning, it is not presented in the results, and that only mentioning that further research is needed may lower the quality and justification of the paper, so additional supplementation is required.
- We are grateful to the reviewer for pointing this out. The following passage has accordingly been added:

Although the present research investigated various non-exceedance probability distributions of typhoon-driven storm surges, it only used a single extreme event in a specified region. As such, its findings may not be applicable to other regions, each of which has its own unique weather conditions, geographic features, and tidal characteristics. Future research should therefore include tidal and environmental data from a range of different regions and various extreme events to test the present study's findings. Also, various natural-hazards indicators and environmental factors such as wind speed, pressure, rainfall, landslides, distance to waterways, and so forth may be useful variables in estimating the exceedance probabilities of typhoons and other natural hazards, and thus be beneficial to risk assessment and mitigation. Also, it should be borne in mind that much of the tidal-gauge data that this study utilised was from the fairly distant past. Thus, in similar future studies, efforts should be made to ensure that such data are reliable, especially in light of climate-change-driven patterns in sea-level behaviour.

Return periods based on various non-exceedance probability models should also be considered in future research, insofar as elaborated return-period estimation can be utilised to improve disaster-relief and emergency-planning efforts. Our comparison of various probability models to find the best fitting distribution models could be adapted to the simulation of time series of the past typhoons, and the collected simulated storm-surge time series then used to estimate typhoons' return periods using bootstrapping of the exceedance data. Potentially, this would provide more exact return periods with confidence intervals. Lastly, future work on return periods should take account of trends in sea-level change, driven by climate change, which already pose a non-negligible risk to coastal buildings and other infrastructure. Advanced statistical methods such as Monte Carlo simulation, as well as deep-learning techniques, could be applied to make typhoon return-period estimates even more accurate.